**Winter 2018 major sudden stratospheric warming impact on midlatitude mesosphere**
**from microwave radiometer measurements**
Yuke Wang[1], Valerii Shulga[1,2], Gennadi Milinevsky[1,3], Aleksey Patoka[2], Oleksandr
Evtushevsky[3], Andrew Klekociuk[4,5], Wei Han[1], Asen Grytsai[3], Dmitry Shulga[2], Valery
Myshenko[2], Oleksandr Antyufeyev[2]
[1]College of Physics, International Center of Future Science, Jilin University, Changchun,
130012, China
[2]Institute of Radio Astronomy, NAS of Ukraine, Kharkiv, 61002, Ukraine
[3]Taras Shevchenko National University of Kyiv, Kyiv, 01601, Ukraine
[4]Antarctica and the Global System, Australian Antarctic Division, Kingston, 7050, Australia
[5]Department of Physics, University of Adelaide, Adelaide, 5005, Australia
*Correspondence to*:
Gennadi Milinevsky (genmilinevsky@gmail.com) and Valerii Shulga
(shulga@rian.kharkov.ua)
**Abstract.** The impact of a major sudden stratospheric warming (SSW) in the Arctic in
February 2018 on the mid-latitude mesosphere is investigated by performing microwave
radiometer measurements of carbon monoxide (CO) and zonal wind above Kharkiv, Ukraine
(50.0°N, 36.3°E). The mesospheric peculiarities of this SSW event were observed using a
recently designed and installed microwave radiometer in East Europe for the first time. Data
from the ERA-Interim and MERRA-2 reanalyses, as well as the Aura Microwave Limb
Sounder measurements, are also used. Microwave observations of the daily CO profiles in
January–March 2018 allowed the retrieval of mesospheric zonal wind at 70–85 km (below the
winter mesopause) over the Kharkiv site. Reversal of the mesospheric westerly from about 10
m s$^{-1}$ to an easterly wind of about –10 m s$^{-1}$ around 10 February was observed. The local
microwave observations at our NH midlatitude site combined with reanalysis data show wide
ranging daily variability in CO, zonal wind and temperature in the mesosphere and stratosphere
during the SSW of 2018. The observed local CO variability can be explained mainly by
horizontal air mass redistribution due to planetary wave activity. Replacement of the CO-rich
polar vortex air by CO-poor air of the surrounding area led to a significant mesospheric CO
decrease over the station during the SSW and fragmentation of the vortex over the station at the
SSW start caused enhanced stratospheric CO at about 30 km. Spectral analysis shows

intensified westward wave 1 throughout the midlatitude upper stratosphere–mesosphere, consistent with other studies of SSWs in the NH winter polar region. The results of microwave measurements of CO and zonal wind in the midlatitude mesosphere at 70–85 km altitudes, which still is not adequately covered by ground-based observations, are useful for improving our understanding of the SSW impacts in this region.

## 1 Introduction

Major sudden stratospheric warming (SSW) events which happen roughly each two years in the North Polar region are produced by strong planetary wave activity according to the model developed by Matsuno (1971) which is supported by numerous observations (Alexander and Shepherd, 2010; Kuttippurath and Nikulin, 2012; Tao et al., 2015). A major SSW event is accompanied by a sharp increase of the stratosphere temperature up to 50 K and the reversal of the zonal wind from climatological westerlies to easterlies over a period of several days (Charlton and Polvani, 2007; Chandran and Collins, 2014; Hu et al., 2014; Tripathi et al., 2016; Butler et al., 2017; Karpechko et al., 2018; Taguchi, 2018; Rao et al., 2018). The primary definition of a SSW event provided by the World Meteorological Organization requires a stratosphere temperature increase and an accompanying zonal wind reversal to easterlies at the 10-hPa pressure level (approximately 30 km altitude) and 60° latitude (WMO, 1978). This definition was broadened and detailed in recent papers (Butler et al., 2015; Butler and Gerber, 2018; Rao et al., 2019). The summarizing paper, where a SSW database is described, was published in Butler et al. (2017). This useful tool (https://www.esrl.noaa.gov/csd/groups/csd8/sswcompendium/) allows analysis of the conditions in the stratosphere, troposphere, and at the surface before, during and after each SSW event representing its evolution, structure, and impact on winter surface climate. The compendium is based on data from six different reanalysis products, covers the 1958–2014 period and includes global daily anomaly fields, full fields, and derived products for each SSW event (Butler et al., 2017).

The source of the SSW is planetary wave activity born in the troposphere that propagates upward through the tropopause to the stratosphere (Matsuno, 1971; Alexander and Shepherd, 2010, Butler et al., 2015). The enhanced wave activity results in the rapid warming of the polar stratosphere and the breakdown of the stratospheric polar vortex (Matsuno, 1971; de la Torre et al., 2012; Chandran and Collins, 2014; Pedatella et al., 2018). The important feature of a SSW event is its impact on lower altitudes, when temperature and wind anomalies descend

downward into the high- and mid-latitude troposphere during the following weeks to month and influence the surface weather (Baldwin and Dunkerton, 2001; Zhou et al., 2002; Butler et al., 2015; Yu et al., 2018). The major SSW events may also impact the atmospheric composition of the whole Northern Hemisphere (NH) stratosphere including mid-latitudes (Solomon et al., 1985; Allen et al., 1999; Tao et al., 2015).

During the SSW, vertical coupling covers not only the troposphere but extends upward to the mesosphere. Mesospheric responses to the SSW are observed as enhancement in planetary wave amplitude, zonal wind reversal and significant air cooling (Shepherd et al., 2014; Zülicke and Becker, 2013; Stray et al., 2015; Zülicke et al., 2018), substantial depletion of the metal layers (Feng et al., 2017; Gardner, 2018), mesosphere-to-stratosphere descent of trace species (Manney et al., 2009; Salmi et al., 2011). The SSW events are also accompanied by the rapid descent of the stratopause into the stratosphere at the SSW onset, following formation of the elevated stratopause in the lower mesosphere and gradual stratopause lowering toward its typical position in the SSW recovery phase (Manney et al., 2009; Chandran et al., 2011; Salmi et al., 2011; Tomikawa et al., 2012; Limpasuvan et al., 2016; Orsolini et al., 2010, 2017). The elevated stratopause events provide an evidence of the coupling between the stratosphere and the mesosphere.

Among the trace gases, the CO molecule is a good tracer of winter polar vortex dynamics in the upper stratosphere and mesosphere due to its long photochemical lifetime (Solomon et al., 1985; Allen et al., 1999; Rinsland et al., 1999, Shepherd et al. 2014). The CO mixing ratio generally increases with height in the upper stratosphere and mesosphere and increases with latitude toward the winter pole. This is due to the mean meridional circulation which transports CO from the source region in the summer hemisphere and tropics to the extratropical winter mesosphere and stratosphere (Shepherd et al., 2014). Therefore, large abundances of CO appear in the winter polar regions under conditions of large-scale planetary wave activity. Downward meridional transport causes descent of CO between the mesosphere and stratosphere and this process is sensitive to planetary wave amplitudes, and particularly the wave amplitude changes that occur during SSWs (Rinsland et al., 1999; Manney et al., 2009; Kvissel et al., 2012). Due to the large scale descent, high CO values of mesospheric origin are observed at stratospheric altitudes down to 25–30 km (Engel et al., 2006; Huret et al., 2006; Funke et al., 2009). At NH mid-latitudes, CO also exhibits significant variability during periods of planetary wave activity associated with SSWs, when the polar vortex splits and displaces off the pole (Solomon et al., 1985; Allen et al., 1999; Funke et al., 2009).

Recent atmospheric models are being extended up to 80–150 km and are used for the study of SSWs (de la Torre et al., 2012; Chandran and Collins, 2014; Shepherd at al., 2014;

Limpasuvan et al., 2016; Newnham et al., 2016). For example, de la Torre et al. (2012) applied
the Whole Atmosphere Community Climate Model (WACCM) and Shepherd at al. (2014) used
the Canadian Middle Atmosphere Model (CMAM) for SSW modeling. The reference wind
profiles for the models are mainly retrieved from observations of the radiation of the
mesospheric ozone molecules, which allow robust measurements at altitudes up to of
approximately 65 km (e.g., Hagen et al., 2018). These data are generally consistent with the
most commonly used reanalysis products. However, there are still insufficient observations of
middle atmospheric winds at altitudes between 60 and 85 km made with a high vertical
resolution to verify atmospheric models and possible long-term trends (Keuer et al., 2007;
Hagen et al., 2018; Rüfenacht et al., 2018). This altitude range, where temperature generally
decreases with height which causes inherent vertical instability, is situated below the winter
mesopause region at 95–100 km (e.g. Xu et al., 2009) and plays a significant role in the mass
and energy exchange between the stratosphere and the mesosphere (Shepherd et al., 2014;
Limpasuvan et al., 2016; Gardner, 2018).
Microwave radiometry is a ground-based technique that can provide vertical profiles of CO,
$H_2O$ and $O_3$ atmospheric gases and wind data in the upper stratosphere and mesosphere
(Rüfenacht et al., 2012; Scheiben et al., 2012; Forkman et al., 2016). The upper stratosphere–
mesosphere zonal winds at the 30–85 km altitude region can be measured using the Doppler
shift between different observation directions in simultaneously measured spectra of transitions
lines of carbon monoxide at 115.3 GHz and ozone $O_3$ at 110.8 GHz (Rüfenacht et al., 2012;
Forkman et al., 2016). Due to high altitude CO residence region, the simultaneous zonal wind
measurements using both $O_3$ and CO provide independent data that extend the wind
measurement from the stratospheric to mesospheric altitudes, respectively (Forkman et al.,
2016; Piddyachiy et al., 2017).
The first ground-based microwave measurements of CO were made in the 1970s and they
confirmed theoretical estimations of the vertical CO profile (Waters et al., 1976; Goldsmith et
al., 1979). Since the 1990s, the ground-based microwave radiometers measuring CO have been
installed in the Northern Hemisphere at high and middle latitudes to provide measurements on
a regular basis. Microwave radiometers are operating in Onsala and Kiruna, Sweden, since
2008. The results are described in Hoffmann et al. (2011) and in Forkman et al. (2012). The
microwave radiometer operated in Bern, Switzerland since 2010 aims to contribute to the
significant gap that exists in the middle atmosphere between 40 and 70 km altitude for wind
data (Rüfenacht et al., 2012). In the Arctic, the $O_3$, $N_2O$, $HNO_3$, and CO spectra were recorded
using the Ground-Based Millimetre-wave Spectrometer GBMS (Muscari et al., 2007; Di Biagio
et al., 2010).

Since 2014, the microwave measuring system for CO observations has been operated in Kharkiv, Ukraine (Piddyachiy et al., 2010; Piddyachiy et al., 2017). Microwave radiometer measurements of CO are used to retrieve mesospheric winds nearby the mesopause region (70–85 km). Methods deriving the wind speed from mesospheric CO measurements are based on the determination of the CO and $O_3$ lines emission Doppler shift (Eriksson et al., 2011; Hagen et al., 2018).

Our observations in February 2018 using the new microwave radiometer at the mid-latitude Kharkiv station have recorded the mesospheric effects of a major SSW. In mid-February 2018, the stratospheric polar vortex in the Arctic splitted into two sister vortices (Fig. 1), the zonal wind reversed in the stratosphere–mesosphere from westerly to easterly and warm air penetrated into the polar cap regions (Rao et al., 2018; Karpechko et al., 2018; Vargin and Kiryushov, 2019). This caused large-scale disturbances in the middle atmosphere of the polar and middle latitudes. The major SSW in 2018 is not yet widely discussed in publications (Rao et al., 2018; Karpechko et al., 2018; Vargin and Kiryushov, 2019) and in this paper, we give a detailed description of the observed mesospheric CO and zonal wind variations.

In Sect. 2, the microwave radiometer and data processing software are briefly described. The SSW event in February 2018 is considered in Sect. 3. The effects of the SSW on mid-latitude mesosphere–stratosphere conditions in the Ukraine longitudinal sector are presented in Sect. 4. Discussion is given in Sect. 5 followed by conclusions in Sect. 6.

**2 Data and methods**

The microwave radiometer data set registered during the 2017/2018 winter campaign in Kharkiv (50.0°N, 36.3°E) is used in this study to investigate local effects of the winter 2018 sudden stratospheric warming on the mesosphere and stratosphere. Since the ground-based microwave measurements are spatially limited by instrument coverage, data on air temperature, zonal wind, geopotential height were used from reanalyses and satellite databases to interpret the CO profile and the zonal wind microwave observations and to describe the SSW effects in the atmosphere of the surrounding mid-latitude region (30–40°E, 48–52°N).

**2.1 Microwave radiometer, method, and midlatitude data description**

The microwave radiometer (MWR) with high sensitivity, installed at Kharkiv, Ukraine, is
designed for continuous observations of the atmospheric CO profiles and zonal wind speed in
the mesosphere using emission lines at 115.3 GHz. The radiometer can continuously provide
vertical profiles up to the mesopause region during day and night, even in cloudy conditions
(Hagen et al., 2018). However, precipitation, such as strong rain or snow, can prevent the
measurements.
The receiver of the radiometer has the double-sideband noise temperature of 250 K at an
ambient temperature of 10°C (Piddyachiy et al., 2010; 2017). The radiometer was tested during
the 2014–2015 period for observation of the CO emission lines in the mesosphere over Kharkiv.
These tests proved the reliability of the receiver system, on which further details are provided
in Piddyachiy et al. (2017). Since 2015, the radiometer has been used for continuous
microwave measurements of CO profiles and mesosphere wind investigations. The first
observations of the atmospheric CO spectral lines over Kharkiv have confirmed seasonal
variations in the CO abundance (Piddyachiy et al., 2017). Operation of the MWR in a double-
sideband mode allows retrieval of wind speed from the Doppler shift of the CO line emission at
the 115.3 GHz. Two methods are used to determine wind speed. Firstly the observed line shape
is fitted by a Voigt profile and the center frequency is determined (Piddyachiy et al., 2017).
Secondly radiative transfer calculations for a horizontally layered atmosphere are used to
determine the wind profiles with the Qpack package, version 1.0.93 (Eriksson et al., 2005;
Eriksson et. al., 2011), which is specifically designed to work with the forward model of the
Atmospheric Radiative Transfer Simulator ARTS (Buehler et al., 2018;
http://www.radiativetransfer.org/). The results obtained by both methods were almost the same
within the error limits. In this paper, both methods were used and provided average values of
the zonal wind speed for altitudes of 70–85 km. The time interval of the data used here was
January 1 – March 31, 2018, which covers the main phases of the SSW 2018 event.


**2.2 Data from other sources**

In this study, daily datasets from ERA-Interim global atmospheric reanalysis of European
Centre for Medium-Range Weather Forecast (ECMWF; Dee et al., 2011) were downloaded
from (https://www.ecmwf.int/en/forecasts/datasets/archive-datasets/reanalysis-datasets/era-
interim) and have been used for comparison with MWR observations. The ERA-Interim data
were used to create temperature and zonal wind velocity profiles and to calculate geopotential
height at the stratospheric pressure levels, in order to compare with the data measured over the

Kharkiv site. Aura Microwave Limb Sounder (MLS) measurements of the air temperature were analyzed as well (Xu et al., 2009; https://mls.jpl.nasa.gov/data/readers.php; see details in the Supplement).

Zonal wave amplitudes in geopotential height were analyzed using the National Oceanic and Atmospheric Administration National Centers for Environmental Prediction, Global Data Assimilation System–Climate Prediction Center (NOAA NCEP GDAS–CPC) data at https://www.cpc.ncep.noaa.gov/products/stratosphere/strat-trop/ and the MERRA-2 data from the National Aeronautics and Space Administration Goddard Space Flight Center, Atmospheric Chemistry and Dynamics Laboratory (NASA GFC ACDL) site at https://acd-ext.gsfc.nasa.gov/Data_services/met/ann_data.html. The detailed description of the data used for analysis is given in the Supplement.

## 3 Northern Hemisphere SSW effects

Descending air masses are observed throughout the mesosphere and stratosphere of the winter polar region (Orsolini et al., 2010; Chandran and Collins, 2014; Limpasuvan et al., 2016; Zülicke et al., 2018). From Aura MLS vertical profiles, a layered descending sequence of alternating cool and warm anomalies over the polar cap was observed in the 2017/2018 winter (Fig. 2a). The SSW event in Fig. 2a is identified by the rapid warming in the stratosphere and cooling in the mesosphere (upward arrow) starting from 10 February 2018 (left vertical line).

This event was preceded by progressively descending warm and cold anomalies that formed in January (black and white dashed arrows, respectively). Oscillations in the intensity of the anomalies indicate that they were formed under the influence of large amplitude planetary waves of zonal wave numbers 1 and 2 (Fig. 2c–2e). From 1 January to 10 February (during 41 days), descending warm anomalies with a velocity $\sim$-850 m·day$^{-1}$ were apparent in the mesosphere and the upper stratosphere (75 to 40 km; black dashed arrow in Fig. 2a). Below the warm anomaly, a cold anomaly descended between the upper and lower stratosphere (45 to 20 km) with velocity $\sim$-600 m·day$^{-1}$ (white dashed arrow in Fig. 2a), while a cold mesospheric anomaly in February–March descended with average velocity $\sim$-750 m·day$^{-1}$ (white dotted arrow in Fig. 2a). Our velocity estimates are similar to those of Salmi et al. (2011) who found

that mesospheric $NO_x$ anomalies during the major SSW 2009 were transported from 80 to 55
km in about 40 days, i.e. with velocity $\sim$-600 m·day$^{-1}$.
The splitting of the polar vortex (Fig. 1) and the zonal wind reversal (Fig. 2b) started at the
time of the wave 2 pulse on 10 February (Fig. 2d and dashed curve in Fig. 2e). Note that this is
close to the SSW timing in Rao et al. (2018) and Vargin and Kiryushov (2019), where the SSW
onset date was 11 February. As seen from Fig. 2c and solid curve in Fig. 2e, increasing wave 1
amplitude contributed to the destabilization of the polar vortex during January–early February
and to temperature and zonal wind oscillations in the mesosphere and stratosphere (Fig. 2a and
2b). These oscillations are usually associated with the propagation of planetary waves in the
stratosphere and mesosphere (Limpasuvan et al., 2016; Rüfenacht et al., 2016). As noted in an
earlier study (Manney et al., 2009; Rao et al., 2018), wave 1 amplitudes were also larger prior
to the SSW in 2009, suggesting a role of preconditioning. During 10–15 February, the easterly
zonal wind anomaly at the stratopause (about 1 hPa, ~50 km) increased to –60 m s$^{-1}$ (Fig. 2b).
At the same time, warming in the polar stratosphere with the largest temperature anomaly of
about 20 K was observed between 25 and 45 km in the same time interval (upward arrow in
Fig. 2a). Both anomaly peaks are close in time to the wave 1 pulse after the SSW start (Fig. 2c
and 2e). The descending negative temperature anomaly in the mesosphere between 50 and 90
km persisted during and after the SSW and reached –15 K (dotted arrow in Fig. 2a).

**4 The local SSW effects over the midlatitude station**

**4.1 CO variability**

Local variability in the conditions of the atmosphere during the microwave measurements in
January–March 2018 at Kharkiv (50°N, 36°E) is shown in Figs. 3–6. The sharp changes
occurred in the 20-day interval from 10 February to 1 March coinciding with the SSW event
2018, as indicated by red vertical lines in Figs. 3, 5 and 6. At this time the polar vortex divided
into two parts producing two smaller vortices over the longitudinal sectors of North America
and Eurasia (Fig. 1). Due to the planetary wave influence (Fig. 2c–2e), the two sub-vortices
shifted zonally and meridionally, so that the SSW effects were observed not only in the polar
region but also in the middle latitudes (Fig. 4).
The CO molecule volume mixing ratio (VMR) near the mesopause at 75–80 km decreased
from 10 ppmv of background level to 4 ppmv on 19–21 February (Fig. 3a), when the sharp
vertical CO gradient at the lower edge of the CO layer near about 6 ppmv increased in height
by about 8 km (between 75 km and 83 km, thick part of the white curve in Fig. 3a). For
comparison, the pre- and post-SSW vertical variations of the 6-ppmv contour were observed in
a range 2–3 km (white curve in Fig. 3a). Moreover, similar variations in the zonal mean 6-
ppmv level are much weaker (yellow curve in Fig. 3e). This indicates that local and regional
mesosphere over the MWR site was disturbed by some source acted during the SSW, which is
identified below. We take here the 6-ppmv contour as a conditional lower edge of the CO layer
since the CO gradients sharply increase from 0.2–0.3 ppmv km$^{-1}$ in a 10-km layer below to
0.6–0.8 ppmv km$^{-1}$ in a 10-km layer above (below and above the white curve in Fig. 3a). The
similar gradient change is characteristic of the mesospheric CO profiles in boreal winter from
ground-based and satellite observations (Fig. 4 in Koo et al., 2017; Fig. 5 in Ryan et al., 2017).
The local mesospheric CO variability from the MWR observations over Kharkiv agrees
with regional one from the MLS data averaged over the adjacent area 47.5–52.5°N, 26–46°E
(Fig. 3b, the white curve for 6 ppmv). However, the zonal mean CO profiles in the same zone
do not show an anomalous decrease of the mesospheric CO during the SSW (yellow curve in
Fig. 3a, 3b and 3e).
The opposite tendency with the stratospheric CO abundance increase is observed from both
regional and zonal mean MLS data shortly after the SSW start (contour 0.1 ppmv in Fig. 3d and
3g, respectively). The CO-rich air of 0.1–0.5 ppmv, which is typical for the lower mesosphere
(Fig. 3c) descended up to about 30 km (Fig. 3d and 3g), far exceeding typical stratospheric CO
mixing ratios on the order of about 0.01–0.02 ppmv (Engel et al., 2006; Huret et al., 2006;
Funke et al. 2009). The CO-rich stratospheric anomaly is close in time to the wave 1 peak on
10–15 February (solid curve in Fig. 2e), that was observed through the stratosphere down to the
30 km altitude (Fig. 2c).
Horizontal distributions of the CO VMR in the Northern Hemisphere at the stratospheric
and mesospheric altitudes in Fig. 4 explain causes of the different CO variability by vertical in
Fig. 3. The dynamical deformation, elongation, and displacements relative to the pole of the
polar vortex lead to temporal shifts in the low and high CO amounts over the MWR site at
Kharkiv (white circle in Fig. 4). The tendency of the planetary wave westward tilt with altitude
(dashed lines in Fig. 4, see also Supplemental Figs. S1 and S2 for more details) also contributes
to relative zonal shift between the stratosphere and the mesosphere of the low/high CO over
Kharkiv.
The observed decrease of the local CO in the mesosphere during the SSW (white curve in
Fig. 3a) is consistent with the regional data from the satellite observations (white curve in Fig.
3b). The decrease is due to the displacement of the CO-rich air to the west relative to Kharkiv
(white circle and contours outlined the CO-rich area in Fig. 4a–4c and 4e–4g). This is a result
of the easterly domination during the SSW that led to placing of the CO-poor air over Kharkiv
with the lowest CO levels on 19–23 February (Fig. 4c and 4g) in correspondence with the
MWR (Fig. 3a) and MLS (Fig. 3b) measurements. Return to the westerly regime in early
March reversed the rotation of the vortex (2–6 March in Fig. 4d and 4h) and caused recovery of
high CO level over Kharkiv (since about 1st of March in Fig. 3a and 3b).
The polar vortex split influenced the local CO change in the middle stratosphere (Fig. 4m–
4o). The low CO level at ~30 km before the SSW start (Fig. 3d) is associated with the relatively
distant location of the CO-rich vortex from Kharkiv (Fig. 4m). The vortex split and easterly
circulation caused displacement of the small vortex fragment with the CO level higher than 0.1
ppmv to Kharkiv just at the SSW start (9–13 February in Fig. 4n) and corresponding sharp CO
increase over the Kharkiv region around 30-km altitude (contour 0.1 ppmv in a few days after
10 February in Fig. 3d). Vertical CO profiles in Fig. 3c and 3d show that downward penetration
of the mesospheric CO-rich air into the startosphere took place around 10 February. As seen
from Fig. 4f, 4j, and 4n, the mesospheric CO-rich air appears to be contained inside the small
sub-vortex over Kharkiv. The large sub-vortex (Fig. 4n and 4o) contributed to the stratospheric
CO increase after 10 February in the zonal mean CO profile near 30 km (Fig. 3g). The two sub-
vortices in Fig. 4n and 4o provided a longer duration of the mesospheric intrusion in the zonal
mean (Fig. 3g) than a short-time influence of the single sub-vortex in regional data (Fig. 3d).
It should be noted that the lower edge of the mid-latitude CO-rich air descended in January
– mid-February (dashed lines in Fig. 3d and 3g) similarly to the temperature anomaly in the
polar region (Fig. 2a). Descent velocity was about -270 and -220 m·day$^{-1}$ in the case of the
regional and zonal mean data, respectively. This is a few times lower than in the vortex region,
nevertheless, it is in the range of the winter descent velocity noted above (Ryan et al., 2018).
Note also that the vortex split in the CO distribution can be identified only in the middle
and upper stratosphere (Fig. 4n and 4o and Fig. S1j and S1k), but not at the stratopause level
(Fig. 4j and 4k) and in the mesosphere (Fig. S2, second and third columns for 9–13 and 19–23
February 2018, respectively).


**4.2 Zonal wind variability**

The reversal of the local zonal wind estimated from the CO measurements at the Kharkiv
MWR site near the mesopause region was observed. The averaged wind velocity in the altitude
range 70–85 km changed between 10 m s$^{-1}$ and –10 m s$^{-1}$ around 10 February (Fig. 5a). Positive
(negative) values are westerly (easterly) wind components. After the active phase of the SSW,
the zonal wind returns to the westerly wind and enhances to 20 m s$^{-1}$ reaching the highest

velocity observed in January–March (Fig. 5a). This zonal wind peak in early March is accompanied by the CO peak at 18 ppmv around 85 km that is also the highest CO abundance over January–March (Fig. 3a). This is closely consistent with the MLS measurements at the 86-km altitude: Kharkiv was located on the 16-ppmv contour in early March (2–6 March in Fig. 4d).

During the SSW event, local zonal wind over the station became easterly between the lower stratosphere and lower mesosphere (–30 m s$^{-1}$ up to –40 m s$^{-1}$, white contours in Fig. 5b). Note that westerly zonal wind at the stratopause level (~50 km) in January 2018 (mid-winter, the pre-SSW conditions) sometimes increased to more than 100 m s$^{-1}$ (black contours in Fig. 5b).

The return of the local westerly wind in the upper mesosphere began in late February (Fig. 5a) and later, in early March, in the lower mesosphere–stratosphere (Fig. 5b). The longer persistence of the westerly anomaly in the stratosphere than at the stratopause level is seen also in the polar region (Fig. 2b). This is a manifestation of the downward migration of the circulation anomalies in the SSW recovery phase, although a near-instantaneous vertical coupling is observed at the SSW start on 10 February (Fig. 2a–2d and Fig. 5).

## 4.3 Temperature changes

The MLS temperature profiles show that high temperature variability over the Kharkiv region concentrated at the stratopause level, particularly before and during the SSW 2018 (Fig. 6). As known, the SSW events are accompanied by polar stratopause descent to 30–40 km, by stratopause breakdown and subsequent reformation at very high altitudes of about 70–80 km (Manney et al., 2009; Chandran et al., 2011; Limpasuvan et al., 2016; Orsolini et al., 2017). The midlatitude stratopause exhibits less sharp, but significant oscillations between 40 and 50 km in January–first half of February 2018 (dotted curve in Fig. 6) and the highest temperature near –5°C after the SSW start on 12–13 February. The short-time stratopause elevation to the lower-mesospheric altitude ~60 km was observed near 20 February, i.e. close in time to the maximum elevation of the 6-ppmv CO level in the mesosphere (Fig. 3a and 3b). Note that the wave 1 and wave 2 (Fig. 2c–2e), and zonal wind (Fig. 5) do not demonstrate strong anomalies this time. The post-SSW stratopause stabilized at the 50-km altitude and warmed from about –20°C to –10°C (Fig. 6b).

Similarly to the CO profile in Fig. 3, the zonal mean temperature variability is much lower above the stratopause than the regional one (Fig. 6b and 6a, respectively). The stratosphere

looks about equally disturbed in both regional and zonal mean characteristics (Fig. 3d and 3g
and Fig. 6a and 6b). This difference may be associated with the influence of the splitted (non-
splitted) polar vortex in the stratosphere (mesosphere). The vortex fragments introduce higher
local/regional and zonal mean variability in the stratosphere; whereas the vortex region is more
uniform in the mesosphere (Fig. 4). That results in the weaker zonal mean variability.
During the SSW, the regional stratospheric temperature in Fig. 6a was warmer by 10–15°C
in comparison with the pre- and post-SSW temperature. This is about two times lower warming
than in the polar region (Fig. 2a) and about three times lower than it is typically observed
during the SSWs (see Section 1). It should be noted that this warm stratospheric anomaly in
Fig. 6a (contour –55°C) rapidly descended between the upper and lower stratosphere (dashed
arrow) in about 10 days. A similar tendency is seen in Fig. 6b from the zonal mean (contour –
55°C) but with a descent within a few days (arrow). So, the SSW start in the midlatitude
stratosphere does not accompany by a near-instantaneous vertical coupling observed in the
polar region (Fig. 2a–2d). Midlatitude stratospheric warming in February 2018 occurred with
increasing time lag between the upper and lower stratosphere.
As is known, upward propagation of the tropospheric planetary waves into the stratosphere
is limited in the easterly zonal wind (Charney and Drazin, 1961). In the changed state of a
zonal flow, the critical line for planetary waves (zero wind line) in the polar region descents in
a few days that looks like downward propagation of an anomaly from above (Matsuno, 1971;
Zhou et al., 2002). Possibly, this process may be delayed in the midlatitude, as seen from Fig.

403    6.



**4.4 Influences of zonal wave 1 and wave 2**

Figure 7 shows time–longitude variations in the MLS temperature anomalies in the Kharkiv
zone 47.5–52.5°N with respect to the mean climatology 2005–2017. The mesospheric and
stratospheric levels during January–March 2018 (Fig. 7a–7c and Fig. 7d and 7e, respectively)
are presented. Dashed lines indicate a sharp change in the direction of zonal migration of the
temperature anomalies from eastward to westward around 10 February. This change coincides
with the reversal of the westerly to easterly at the SSW start (Fig. 2b and Fig. 5). Alternating
sequences of the positive and negative anomalies in Fig. 7 indicate the planetary wave ridges
and troughs migrating along the midlatitude zone.

In the lower–middle stratosphere (22 km in Fig. 7e, 24 and 30 km in Fig. S3h and S3i), the change in the anomaly migration direction is not as pronounced as at the upper levels. The slowly westward migrating positive anomaly is a wave 1 ridge that dominates in the eastern longitudes (black solid line in Fig. 7e and Fig. S3h–S3j). Note that the Kharkiv longitude 36°E (white line in Fig. 7 and Fig. S3) remains out of the wave 1 ridge during January–March. Wave 1 ridge weakens with altitude and wave 1 trough becomes deeper in the western upper stratosphere (Fig. 7d and Fig. S3e–S3g). The vertical wave transformation is accompanied by a westward tilt with altitude seen from the sequential westward shift of both wave 1 ridge and wave 1 trough (solid and dashed lines, respectively, in Fig. S3). This tendency is consistent with the upward propagation of the planetary waves.

Migrating anomalies weaken rapidly after the SSW (to the right of the red vertical line on 1st of March in Fig. 7) as a result of the general decrease in wave activity (Fig. 2e). The results of Fig. 7 and Fig. S3 suggest modification of the zonal wave spectra in time and altitude and Fig. 8 and Fig. 9 present the zonal wave spectra in the lower–middle stratosphere and upper stratosphere–mesosphere, respectively. Figure 8 shows spectra at three levels: 23, 27 and 31 km (lower, middle and upper panel, respectively).

It is seen that short periods <5 days are not statistically significant at these altitudes. Eastward wave 1 exhibit a maximum variance at 10–30 day periods (red curve in Fig. 8d–8f). Westward wave 1 and eastward wave 2 (black and blue curves in Fig. 8d–8f) do not show clear periodicity peak and tend to be more intense at the longest periods, i.e. to be quasi-stationary. This is confirmed by spectra in Fig. 8g–8i. Westward wave 1 apparent from Fig. 8a–8c (black solid line along the wave ridge) is of highest spectral power in Fig. 8d–8f (black curve) and in Fig. 8g–8i (the black vertical line at wave number –1).

To examine the wave spectrum difference in the upper stratosphere–mesosphere before and after the SSW start that is suggested by Fig. 7, the two 40-day time intervals are compared in Fig. 9. These are 20 December–10 February and 10 February–31 March for the intervals of pre- and post SSW initial date, respectively. It is seen from Fig. 9a–9e (Fig. 9f–9j) that eastward (westward) wave 1 demonstrates maximum spectral signal before (after) the SSW start. Transition from eastward to westward propagated wave 1 is seen also from the wave number spectra in Fig. 9k–9o and Fig. 9p–9t), respectively. If the short and long periods (<5 days and >5 days) are present in the first interval, then the periods longer than 10 days dominate in the second interval (Fig. 9k–9o and Fig. 9p–9t, respectively).

The role of wave 1 and wave 2 in the SSW preconditioning and development is known from many studies (Matsuno, 1971; Charlton et al., 2007; Manney et al., 2009; Yuan et al., 2012; Limpasuvan et al., 2016; Rao et al., 2018). Our spectral analysis (Fig. 8 and Fig. 9)

reveals the changes in the wave spectra associated with the SSW onset and their altitudinal
dependence.

**5 Discussion**

The observations of the major SSW effects in February 2018 in the NH midlatitude mesosphere
by microwave radiometer at the Kharkiv site, Northern Ukraine (50.0°N, 36.3°E), have been
provided. The CO altitude profiles in the mesosphere have been measured by the MWR with
one-day time resolution. Using the CO molecule as a tracer, the wind speed has been retrieved
from the Doppler shift of the CO 115.3 GHz emission and the mesospheric winds reverse from
westerly to easterly below the winter mesopause region (70–85 km) has been detected. A few
ground-based observations in the mesosphere by the same method have been undertaken at
midlatitudes (Sect. 1). The zonal wind and CO profile variability during the major SSW were
compared with the daily zonal wind, temperature, zonal wave 1/wave 2 and geopotential height
datasets from the MLS data, the ERA-Interim, and MERRA-2 reanalyses. The SSW started
with the polar vortex split around 10 February (Fig. 1), zonal wind reverse in the mesosphere
and stratosphere (Fig. 2b and Fig. 5) and enhanced stratosphere warming and mesosphere
cooling (Fig. 2a).
Among the most striking SSW manifestations over the midlatitude station in February
2018, there were (i) zonal wind reversal throughout the mesosphere–stratosphere, (ii)
oscillations in the vertical profiles of CO, zonal wind and temperature, (iii) descent of the
stratospheric CO and temperature anomalies on the time scale of days to months, (iv) change
from the eastward to westward wave 1 around the starting date of the SSW and (v) strong
mesospheric CO and westerly peaks at the start of the SSW recovery phase. The midlatitude
SSW effects are known from many event analyses and in most cases they are associated with
zonal asymmetry and polar vortex split and displacements relative to the pole (Solomon et al.,
1985; Allen et al., 1999; Yuan et al., 2012; Chandran and Collins, 2014). Our results show that
the local midlatitude atmosphere variability in the SSW 2018 combines both the large-scale
changes in the zonal circulation and temperature typical for the SSWs and the altitude-
dependent planetary wave patterns and their evolution in the individual vortex split event.


**5.1 Wave patterns and CO level**

As noted in Sect. 1, CO abundance in the extratropical mesosphere increases with latitude toward the winter pole due to meridional transport. CO accumulation results in the formation of the CO layer with the sharp vertical gradient at its lower edge (Solomon et al., 1985; Shepherd et al., 2014). The horizontal CO gradient at the polar vortex edge also exists and the vortex split and displacement of the pole associated with the SSW cause significant CO variability at the NH midlatitudes (Solomon et al., 1985; Allen et al., 1999; Funke et al., 2009; Shepherd et al., 2014).

In Sect. 4a, based on the MWR observations, we have defined the lower CO edge at 6 ppmv and this edge uplifted during the SSW by about 8 km (between 75 km and 83 km, thick part of the white curve in Fig. 3a). This uplifting noticeably stands out against the pre- and post-SSW variations of the 6-ppmv level occurring within 2–3 km (Fig. 4a). The MLS CO measurements show similar variations in the 6-ppmv level over the Kharkiv region (white curve in Fig. 3b) and their absence in the corresponding zonal mean (yellow curve in Fig. 3a, 3b, and 3e).

Mesospheric CO profile uplifting is usually associated with the stratopause elevation during the SSW, when air, poor in CO, enters the mesospheric CO layer from below (Kvissel et al., 2012; Shepherd et al., 2014). Similar ascending motions in the stratopause and mesopause regions were observed in the 2013 SSW from nitric oxide (NO) and showed that the NO contours deflected upwards throughout the mesosphere (Orsolini et al., 2017). Our analysis reveals that the local CO profile variations during the SSW 2018 were closely associated with the changes in the planetary wave patterns in the mesosphere.

The MLS CO distribution demonstrates how deformation, elongation (wave 2 effect) and rotation of the CO-rich polar area influence the local CO level over Kharkiv (white circle with respect to the CO contours in Fig. 4a–4h and Fig. S1). The highest elevation of the 6-ppmv CO level in Fig. 3a and 3b corresponds to the lowest CO level over Kharkiv on 19–23 February, when the most distant displacement of the CO contours 16 ppmv and 6 ppmv off the Kharkiv location was observed (Fig. 4c and 4g, respectively; see also the third column in Fig. S1). As known, the strong vertical CO gradient in the winter mesosphere is found at the higher altitudes in the tropics than in the extratropics (Solomon et al., 1985; Allen et al., 1999; Garcia et al., 2014). Then, poleward displacement of the low-latitude air masses is accompanied by the CO abundance decrease and vertical CO gradient elevation at the middle latitudes, as it is observed in Fig. 3a and 3b. A similar effect related to the wave 1 influence was observed during the 2003–2004 Arctic warming (Funke et al., 2009): the vortex has shifted from the pole toward the western sector and mid-latitude air poor in CO filled the eastern sector (0–90°E) over 50–80°N and even over the pole.

The results of Fig. 4 and Fig. S1 show that meridional displacements of the low-latitude, CO-poor mesospheric air to the Kharkiv region occurred under the planetary wave influence and caused the local CO profile variations in the SSW 2018 (Fig. 3a and 3b). These results, thus, confirm that latitudinal displacements due to wave effects may dramatically affect the local densities of the atmospheric species (Solomon et al., 1985). Figure 6a demonstrates that the local stratopause elevation in February 2018 to about 60 km was relatively small in comparison with the elevation that is characteristic for the polar region, up to 70–80 km (Chandran et al., 2011; Tomikawa et al., 2012; Limpasuvan et al., 2016; Orsolini et al., 2010, 2017). No significant stratopause elevation was observed in the zonal mean for 47.5–52.5°N (Fig. 6b). Therefore, the meridional (poleward) and zonal displacements of the CO-rich air masses enclosed within the polar vortex (Solomon et al., 1985; Allen et al., 1999; Funke et al., 2009) rather than stratopause elevation (Kvissel et al., 2012; Shepherd et al., 2014) may be dominant cause of the CO profile uplift observed in the NH midlatitudes during the SSW 2018.

In March 2018, after the SSW, vertical CO profile has been re-established (Fig. 3a and 3b) according to the recovery phase following the SSW (Shepherd et al., 2014; Limpasuvan et al., 2016). In the MWR data, the SSW recovery phase in the mesosphere in early March started with the short-term but anomalously high peaks in the local CO (Fig. 3a) and westerly wind (Fig. 5a). These peaks reached the highest values in daily variations of CO and zonal wind over the three months of the observations (January–March). By analogy with the low-CO episode in February discussed above, the high-CO peak in early March 2018 caused by change in the vortex shape and return of the CO-rich vortex edge region to the Kharkiv location (compare 2–6 March in Fig. 4d and 4h with 19–23 February in Fig. 4c and 4g; see also the same dates in Fig. S2).

Wind measurements using the CO layer provides a further means to evaluate the validity of the modeled winds. Furthermore, by combining the measurements with ray tracing of gravity wave propagation (e.g. Kogure et al., 2018), this type of measurement may provide particular insights into wave-mean flow interactions, particularly where local temperature inversions alter gravity wave filtering (Hocke et al., 2018; Fritts et al., 2018).

**5.2 Descent of the midlatitude stratospheric anomalies**

Alternating altitudinal sequence of warm and cool anomalies progressively descended through
the mesosphere and stratosphere of the polar region was observed in January–March 2018 (Fig.
2a) in consistency with many observations (Zhou et al., 2002; Orsolini et al., 2010; Shepherd et
al., 2014; de Wit et al., 2014; Zülicke et al., 2018). The warm anomaly sharply intensified in
the stratosphere between 20 and 50 km with simultaneous strong cooling in the mesosphere in
the active phase of SSW since 10 February (vertical arrow in Fig. 2a). Unlike this, the
midlatitude temperature anomalies do not show the similar vertical arrangement and regular
descent with respect to the same mean climatology 2005–2017 (Fig. S4).
During the SSW of 2018, the upper (lower) stratosphere over the Kharkiv region was cooler
(warmer) up to 20°C (10°C) than climatological mean with stepwise descent relative to the pre-
SSW one (Fig. S4a). However, excluding unstable anomalies at different altitudes, the air
temperature through the mesosphere and stratosphere was close to the climatology during most
of the time in January–March 2018 (light blue in Fig. S4a). The zonal mean temperature
anomalies show steady warming of the air in the stratosphere and lower mesosphere and
distinct tendency for the anomaly to descend between about 40 km and 20 km during the SSW
(20 days, $\sim -1$ km·day$^{-1}$). It could be concluded that the temperature anomaly profile observed
in the NH midlatitudes may vary in time depending on the observing location and individual
SSW event and, thus, differ from climatologically warm (cold) stratospheric (mesospheric)
anomaly typical for the SSWs in the NH polar region (e.g. Chandran and Collins, 2014; their
Fig. 1g).
The CO profiles in Fig. 3 demonstrate opposite tendencies in the vertical shift of the CO-
rich air in the NH midlatitudes. The CO descent in the stratosphere occurred during January–
February with velocities of about 270 and 220 m·day$^{-1}$ in a case of the regional and zonal mean
data, respectively (Fig. 3d and 3g). In general, this is in a range of the winter descent velocities
observed in the polar vortex (Funke et al., 2009; Salmi et al., 2011; Ryan et al., 2018),
however, a few times lower than in the polar vortex in the winter 2017–2018 (Fig. 2a). The
deepest penetration of the mesospheric CO levels (0.1–0.5 ppmv) to ~30 km was observed
immediately after the SSW onset (Fig. 3d and 3g). Although this coincides with the peaks in
the wave 1 and wave 2 amplitudes (Fig. 2e), the main reason in the CO increase in the
stratosphere over Kharkiv is the location of the small sub-vortex of the splitted polar vortex (9–
13 February, Fig. 4n).
The MLS CO maps in Fig. 4 show that the high CO amount is concentrated inside the polar
vortex and its fragments after splitting. This is a result of meridional and downward transport of
CO that is strongest in the winter polar vortex (Rinsland et al., 1999; Manney et al., 2009;
Kvissel et al., 2012; Shepherd et al., 2014). Before (4–8 February), during (19–23 February)
and after (2–6 March) the SSW, Kharkiv was outside the stratospheric vortex/sub-vortices edge
(Fig. 4m, 4o and 4p, respectively) and the CO amount was at low level typical for the
midlatitude stratosphere (of about 0.01–0.02 ppmv; Engel et al., 2006; Huret et al., 2006; Funke
et al. 2009). Descent of the 0.1-ppmv contour marked by dashed lines in Fig. 3d and 3g is
observed due to the episodic shift of the vortex edge toward the Kharkiv region or to the
corresponding zone 47.5–52.5°N, respectively.
Figure 4 demonstrates that the CO amount inside the polar vortex or its fragments is much
higher than in the surrounding area not only in the mesosphere but also in the stratosphere. This
leads to the possibility of the enhanced CO appearance even in the stratosphere at about 25–30
km (Engel et al., 2006; Huret et al., 2006; Funke et al., 2009). By analogy, the vortex edge shift
beyond the Kharkiv region (Fig 4c and 4g) resulted in lowering of the regional CO mixing
ratios in the mesosphere consistent to both ground-based and satellite observations (Fig. 3a and
3b, respectively). Meridional structure of the mesospheric CO (Sect. 1) provided the uplift of
the 6-ppmv level during the SSW relative to pre- and post-SSW levels (Fig. 3a and 3b).

**5.3 Wave spectrum changes**

As known, amplified wave 1 and wave 2 are dominant zonal wave numbers in the stratosphere
and mesosphere that precede the SSW and cause zonal wind reversal and polar vortex
displacement off the pole (wave 1) or vortex split (wave 2) at the start and during the SSW
(Matsuno, 1971; Charlton et al., 2007; Manney et al., 2009; Yuan et al., 2012; Limpasuvan et
al., 2016). Variations in the wave amplitudes (Fig. 2e) are a possible cause of the oscillations in
CO, zonal wind and temperature described in Sect. 4. In addition to variability in the anomaly
intensity, the character of the zonal circulation is under the wave influence on the different
SSW phase (Sect. 4.2). Particularly, the spectral composition of the waves is reflected in the
temperature anomaly zonal migration (Sect. 4.4) to which less attention was given in the earlier
studies. Clear change from eastward to westward anomaly propagation is seen in the upper
stratosphere–mesosphere at the SSW initial date, 10 February 2018 (Fig. 7 and Fig. S3) and it
coincides with the zonal wind reversal from westerly to easterly (Fig. 2b and Fig. 5).
Corresponding changes occurred in the wave spectra (Fig. 9) with prevailing eastward
(westward) wave 1 before (after) 10 February.
The simulations made by Limpasuvan et al. (2016) show that the westward propagating
planetary wave 1 forcing dominates above 70 km in the winter hemisphere with the SSW onset.
Since upward planetary wave propagation is limited in the easterly zonal flow (Charney and
Drazin, 1961), the presence of in situ forced planetary waves around the SSW onset due to the
jet instability in the underlying polar mesosphere is discussed (Limpasuvan et al., 2016, and
references herein). Limpasuvan et al. (2016) have shown that spectral power of the westward
wave 1 increases around the SSW onset also in the 40–60 km layer (their Fig. 10b) and this
effect may be caused by unstable westward polar jet below 80 km. The results of Section 4.4
(Fig. 9) suggest that some kind of instability and westward wave forcing down to the upper
stratosphere is possible in the midlatitudes. This possibility needs to be examined in the
simulations.


**6 Conclusions**

The impact of a major sudden stratospheric warming (SSW) in February 2018 on the mid-
latitude mesosphere was investigated using microwave radiometer measurements in Kharkiv,
Ukraine (50.0°N, 36.3°E). The zonal wind reversal has been revealed below the winter
mesopause region at 70–85 km altitudes during the SSW using the CO profiles. The reverse of
the mesospheric westerly from about 10 m s$^{-1}$ to easterly wind about –10 m s$^{-1}$ around 10
February has been documented. The data from the ERA-Interim and MERRA-2 reanalyses and
the Aura MLS temperature profiles have been used for the analysis of stratosphere–mesosphere
behavior under the SSW conditions. Our local microwave observations in the NH midlatitude
combined with the reanalysis data show wide ranges of daily variability in CO, zonal wind and
temperature in the mesosphere and stratosphere during the SSW 2018.
Among the most striking SSW manifestations over the midlatitude station in February
2018, there were (i) zonal wind reversal throughout the mesosphere–stratosphere, (ii)
oscillations in the vertical profiles of CO, zonal wind and temperature, (iii) descent of the
stratospheric CO and temperature anomalies on the time scale of days to months, (iv) wave 2
peak at the vortex split date and change from the eastward to westward wave 1 during the SSW
and (v) strong mesospheric CO and westerly peaks at the start of the SSW recovery phase.
Generally, the midlatitude SSW effects are known from many event analyses and in most cases
they are associated with zonal asymmetry and polar vortex split and displacements relative to
the pole (Solomon et al., 1985; Allen et al., 1999; Yuan et al., 2012; Chandran and Collins,
2014). From our results, the local midlatitude atmosphere variability in the SSW 2018 combine
both the large-scale changes in the zonal circulation and temperature typical for the SSWs and

local evolution of the altitude-dependent planetary wave patterns in the individual vortex split event.

The observed local CO variability can be explained mainly by horizontal air mass redistribution due to planetary wave activity with the replacement of the CO-rich air by CO-poor air and vice versa, in agreement with other studies. The MLS CO fields show that the CO-rich air masses are enclosed within the polar vortex. Horizontal (meridional and zonal) displacements of the edge of the vortex or vortex fragments relative to the ground-based midlatitude station may be a dominant cause of the observed CO profile variations during the SSW 2018. The small sub-vortex located over the station at the SSW start caused the appearance of the enhanced CO level not only in the mesosphere but also in the stratosphere at about 30 km. This indicates that the polar vortex contains the CO-rich air masses with much higher CO amount that in the surrounding area and this takes place over the stratosphere–mesosphere altitude range.

Microwave observations show that sharp altitudinal CO gradient below the mesopause could be used to define the lower edge of the CO layer and to evaluate oscillation and significant elevation of the lower CO edge during the SSW and its trend on a seasonal time scale. The presented results of microwave measurements of CO and zonal wind in the midlatitude mesosphere at 70–85 km altitudes, which is still not adequately covered by ground-based observations (Hagen et al., 2018; Rüfenacht et al., 2018), are suitable for evaluating and potentially improving atmospheric models. Simulations show that planetary wave forcing by westward propagating wave 1 dominates between 40 and 80 km in the winter polar region during the SSW (Limpasuvan et al., 2016). Our spectral analysis reveals that the westward wave 1 during the SSW 2018 is a dominant wave component through the midlatitude upper stratosphere–mesosphere. Instability of the westward polar jet suggested in previous studies (e.g. Limpasuvan et al., 2016) should be analyzed in the context of the westward wave 1 generation in the midlatitude upper stratosphere–mesosphere.

Our observation of variability of the CO layer during the SSW deserves further study, particularly in relation to the implications for modelling of wave dynamics and vertical coupling (Ern et al., 2016; Martineau et al., 2018) and chemical processes (Garcia et al., 2014) in the mesosphere.

*Conflict of Interest.* The authors declare that the research was conducted in the absence of any commercial or financial relationships that could be construed as a potential conflict of interest.

*Author contributions.* GM coordinated and led the efforts for this manuscript. VS initiated the microwave measurements during the SSW event in Kharkiv. VS, DS, VM and AA developed equipment and provided microwave measurements with data processing by AP and DS. GM, VS, YW, OE, AK, and AG analyzed the results and provided interpretation. GM, OE, AK, VS, and WH wrote the paper with input from all authors.

*Acknowledgments.* This work was supported in part by the Institute of Radio Astronomy of the National Academy of Sciences of Ukraine; by Taras Shevchenko National University of Kyiv, project 19BF051-08; by the College of Physics, International Center of Future Science, Jilin University, China. The microwave radiometer data have been processed using ARTS and Qpack software packages (http://www.radiativetransfer.org/). Daily datasets from ERA-Interim reanalysis of European Centre for Medium-Range Weather Forecast (ECMWF) were downloaded from https://www.ecmwf.int/en/forecasts/datasets/archive-datasets/reanalysis-datasets/era-interim. The Aura Microwave Limb Sounder (MLS) measurements of air temperature and CO were obtained from https://mls.jpl.nasa.gov/data/readers.php. Zonal waves were analyzed using the National Oceanic and Atmospheric Administration National Centers for Environmental Prediction, Global Data Assimilation System–Climate Prediction Center (NOAA NCEP GDAS–CPC) data at https://www.cpc.ncep.noaa.gov/products/stratosphere/strat-trop/ and the MERRA-2 data from the National Aeronautics and Space Administration Goddard Space Flight Center, Atmospheric Chemistry and Dynamics Laboratory (NASA GFC ACDL) site at https://acd-ext.gsfc.nasa.gov/Data_services/met/ann_data.html. Authors thank the two anonymous reviewers for their valuable comments and useful suggestions.

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

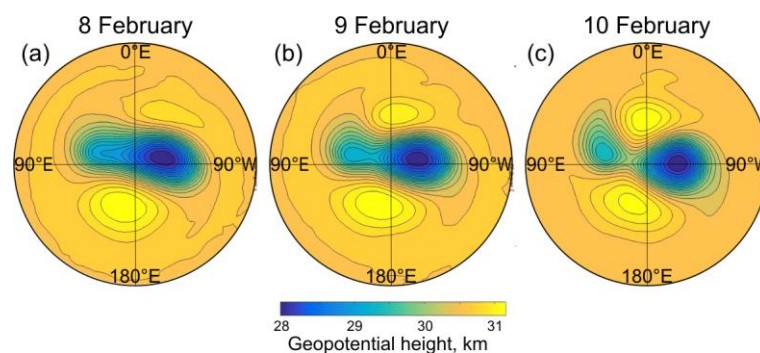



**Figure 1.** The polar vortex split at the 10-hPa pressure level during the SSW event in February
2018. Geopotential heights are calculated from ERA-Interim reanalysis data.


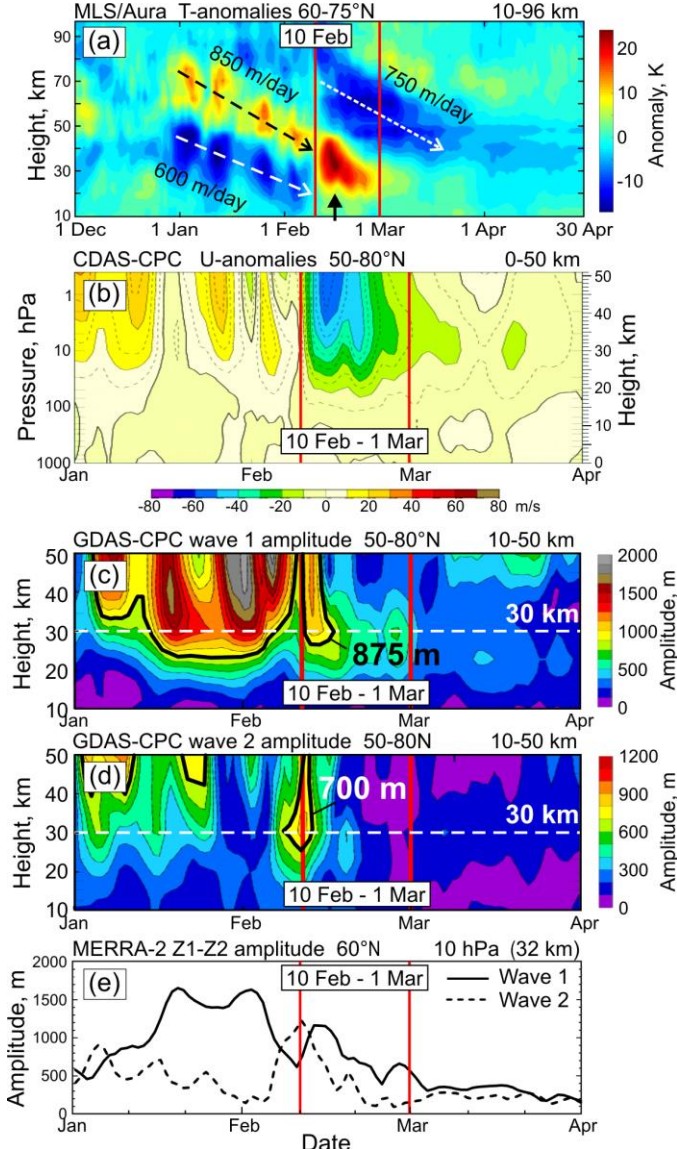



**Figure 2.** The development of the SSW in 2018 from the vertical profiles of (a) Aura MLS temperature anomalies in December 2017–April 2018 at polar zone 60–75°N (with respect to the mean climatology 2005–2017), (b) zonal mean zonal wind anomalies, (c) wave 1 and (d) wave 2 amplitudes in geopotential height in January–March by NOAA NCEP GDAS-CPC data (climatology 1981–2010). (e) zonal wave 1 and wave 2 amplitudes in geopotential height at 10 hPa, 60°N, by the MERRA-2 time series from the NASA GFC ACDL data. The SSW-related anomalous variability between 10 February and 1 March 2018 is bounded by red vertical lines.




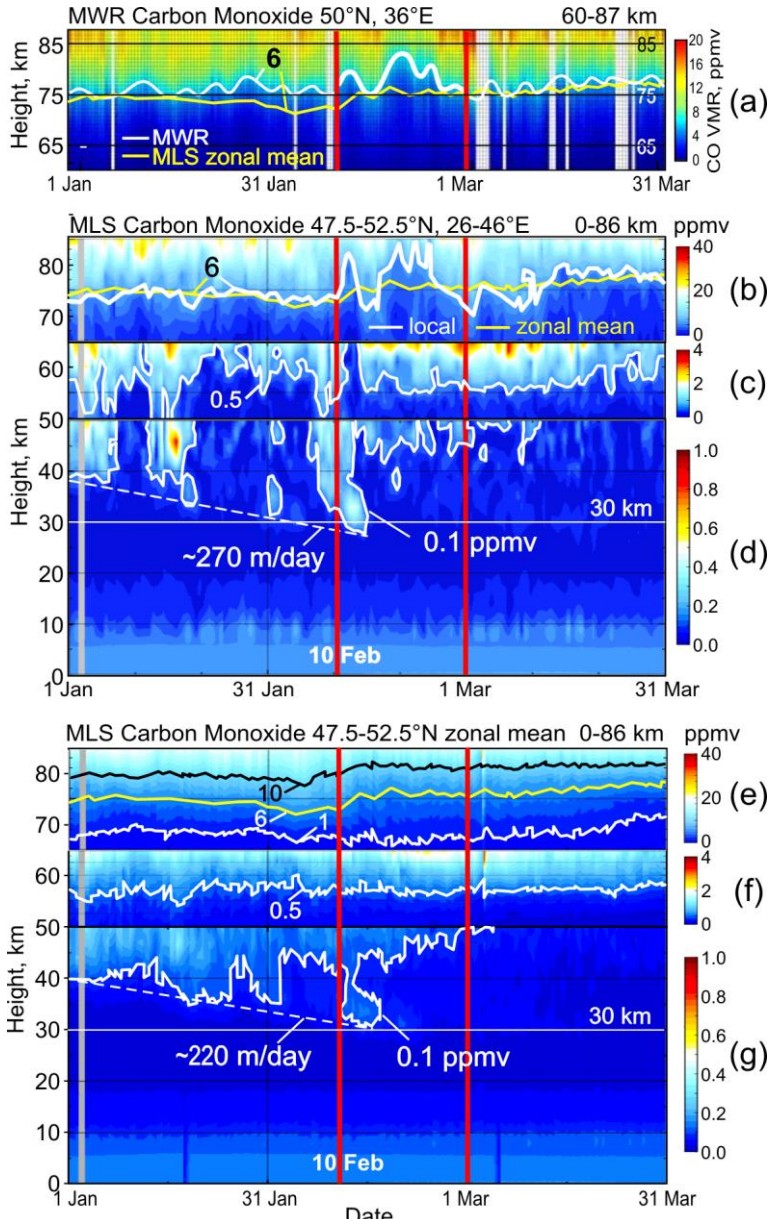


**Figure 3.** (a) Mesospheric CO profile from microwave measurements over Kharkiv averaged
in altitude range 70–85 km, and vertical CO profile from the MLS measurements averaged over
latitudes 47.5–52.5°N and longitudes (b)–(d) 26–46°E centered at the Kharkiv MWR site
(50°N, 36°E) and (e)–(g) 0–360°E for zonal mean. Selected CO levels are highlighted by white,
black and yellow contours (see text for details). Data for January–March 2018 are presented
and time interval of significant variations in the atmosphere parameters due to the SSW event
(from 10 February to 1 March 2018) is bounded by red vertical lines.




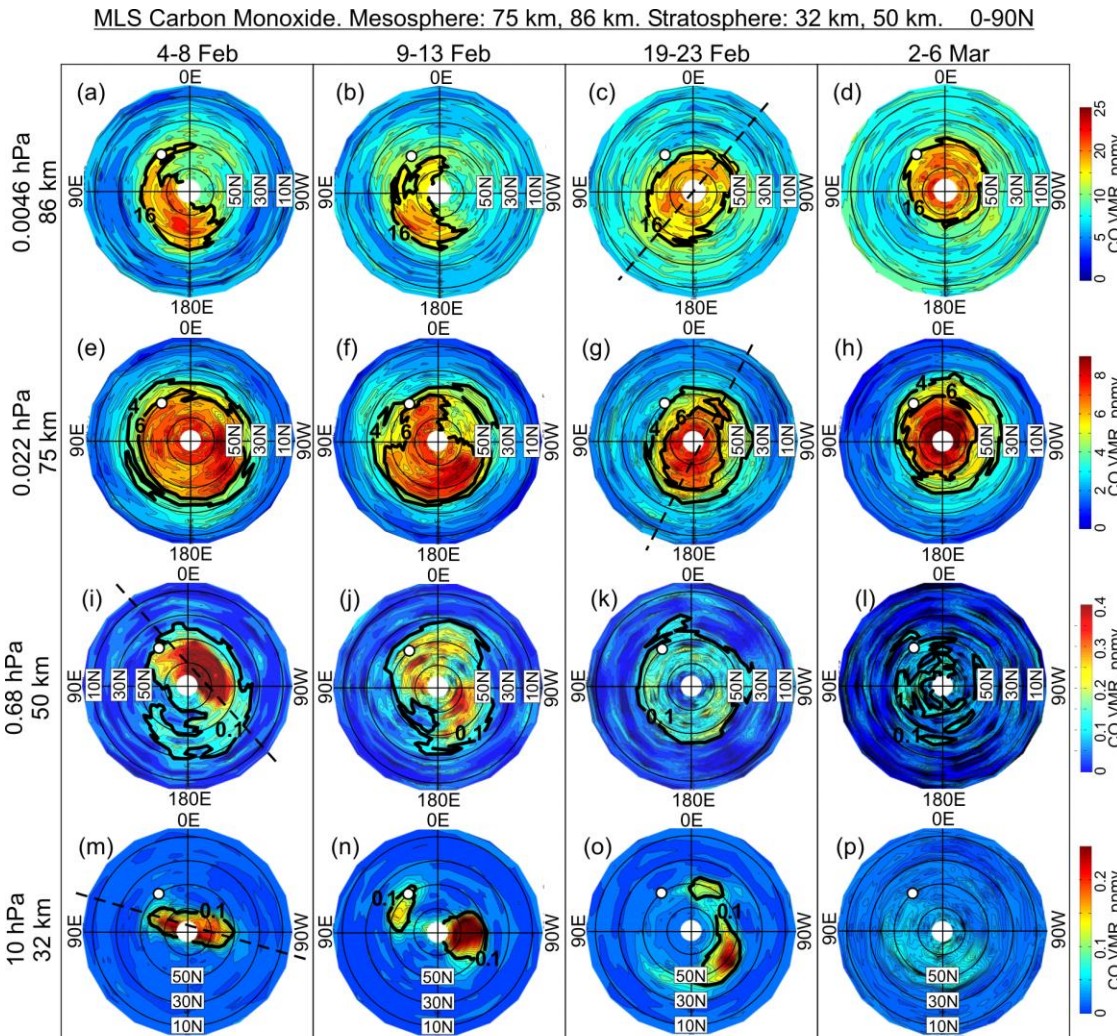



**Figure 4.** The 5-day mean CO field over the NH (0–90°N) from the MLS measurements at the two mesospheric (75 km and 86 km) and stratospheric (32 km and 50 km) levels before (4–8 February), during (9–13 and 19–23 February) and after (2–6 March) the SSW 2018. White circle shows location of the MWR site Kharkiv relatively the high/low CO amounts marked off by the black contours. Dashed lines indicate clockwise rotation of the elongated polar vortex with altitude as manifestation of upward propagation of planetary waves with their westward tilt with altitude.







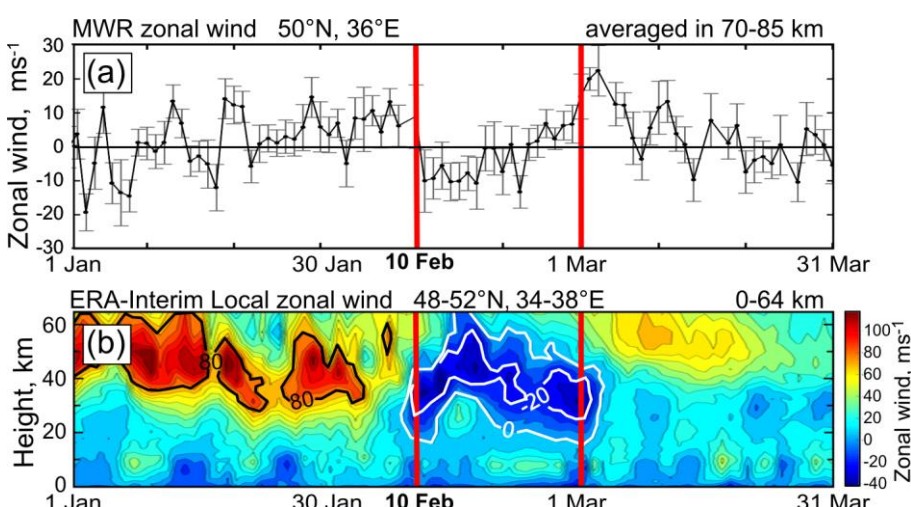



**Figure 5.** (a) Mesospheric zonal wind microwave measurements over Kharkiv (averaged in
altitude range 70–85 km, vertical bars are standard deviations) compared to (b) time-altitude
local zonal wind from the ERA-Interim reanalysis data averaged over latitudes 48–52°N and
longitudes 34–38°E (centered at the Kharkiv microwave radiometer site, 50°N, 36°E ). Time
interval of significant variations in the atmosphere parameters due to the SSW event (from 10
February to 1 March, 2018) is bounded by red vertical lines.





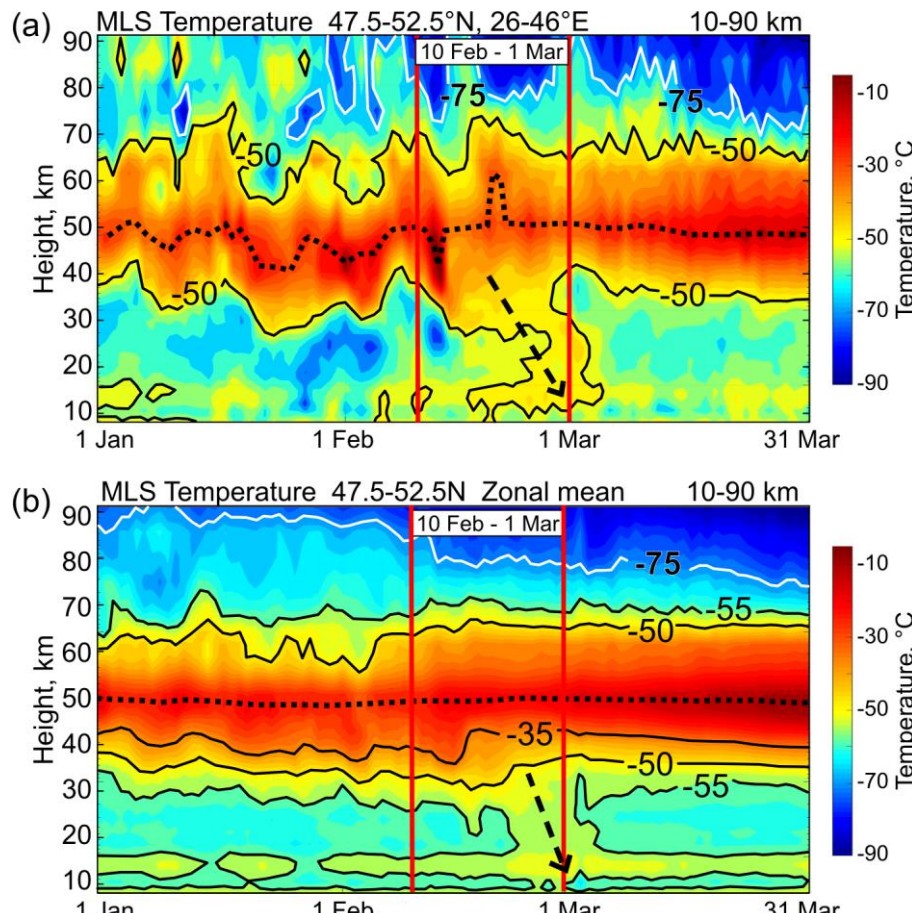



**Figure 6.** MLS temperature profiles (a) over the Kharkiv region and (b) zonal average in the
zone 47.5–52.5°N. Dashed arrows indicate downward warming.





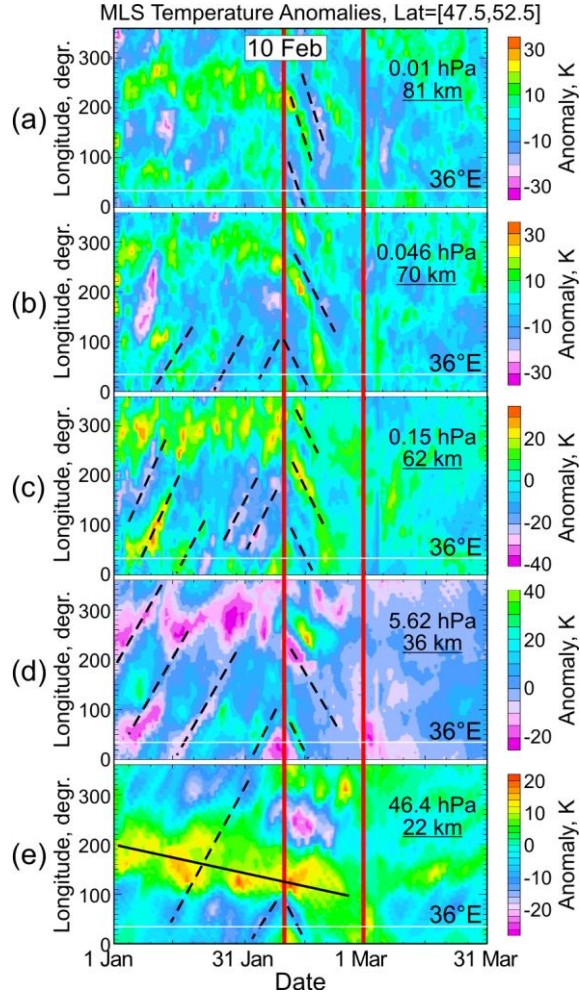



**Figure 7.** Time–longitude variations of the MLS temperature anomalies in the Kharkiv zone
47.5–52.5°N with respect to the mean climatology 2005–2017 during January–March 2018.
Dashed lines show change of the zonal anomaly propagation from eastward to westward near
10 February, at the start of the SSW 2018.





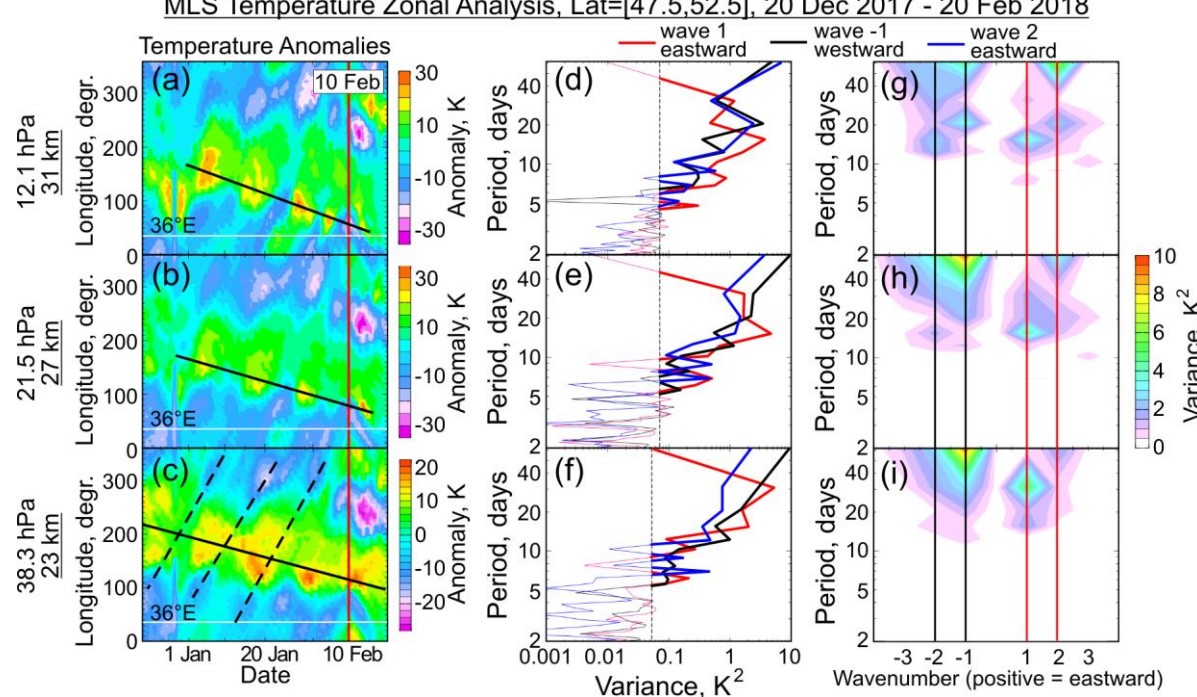



**Figure 8.** (left) As in Fig. 7, but for the zonal temperature anomalies in the lower–middle
stratosphere at 23, 27 and 31 km (lower, middle and upper panels, respectively) during 20
December 2017 – 20 February 2018; (middle) wave 1 and wave 2 periods versus variance and
(right) wave number spectra for the corresponding altitudes. Dashed line in the middle column
marks the 95% confidence limit and bold curves highlight the wavenumber variance exceeding
this limit.



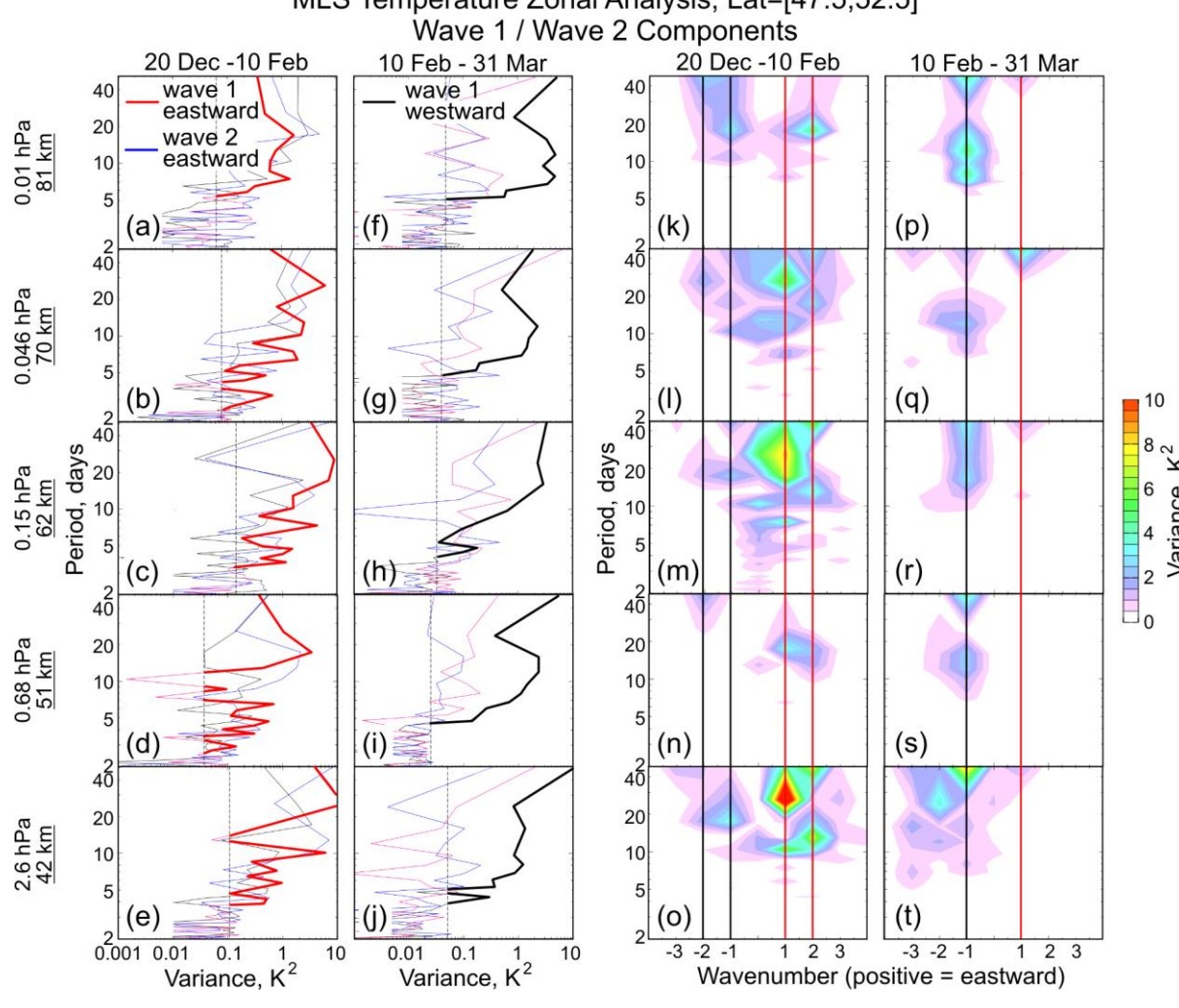

**Figure 9.** The spectral analysis of the zonal temperature anomalies as in Fig. 8 (middle and right) but for the upper stratosphere–mesosphere: (a–e, k–o) before and (f–j, p–t) after the SSW start on 10 February 2018. Red and black lines indicate the eastward and westward propagating wavenumbers, respectively. Bold curves to the right of dashed line in (a–j) and spectra in (k–t) show the wavenumber variance exceeding the 95% confidence limit.