# Peer review of "Winter 2018 major sudden stratospheric warming impact on midlatitude mesosphere"

_Atmospheric Chemistry and Physics, 2018_

## Referee Comment (RC1) · Anonymous Referee #1 · 7 Feb 2019

GENERAL COMMENTS

The topic of ground-based mesospheric carbon dioxide observations is interesting and fits into the scope of ACP. The local microwave radiometer observations over Kharkiv during the major warming event in 2018 are valuable, but they need to be validated and explained adequately. In this sense I suggest the authors to add stratospheric carbon dioxide and mesospheric temperature data for a solid discussion of horizontal and vertical transports. If this information is provided in a concentrated form during a *major revision*, the paper would gain further value.

SPECIFIC COMMENTS

1) Validation: A validation of the reported CO observations is required. For this purpose satellite observations could be used 1) to compare the CO in the 60-90 km range and 2) to extend the profiles will into the stratosphere. This would allow the qualification of downward transport during the 2018 major warming event in comparison with other events. Note, that the shown changes remain in the 70-80 km range while Funke et al. (2009) reports on effects down to 30 km for the 2004 major warming event. Such a three-dimensional picture would allow to place the local observations into a global context.

2) Explanation: The present explanation of the observed mesospheric CO profile in terms of horizontal and vertical transports is highly speculative. It uses analysis data for the stratosphere which does not directly imply a clear picture of the mesosphere. So you can not use the presence of stratospheric planetary waves to explain oscillations in the mesosphere on the daily basis. For such a link mesospheric data are required, which exist with satellites. In this sense, you give some information on MLS-derived temperatures in figures 2.a and 7, but the latter is very noisy and not very helpful. The presentation of maps at selected levels would considerably help the interpretation of local CO and T observations in terms of three-dimensional transports.

3) Concentration: In some points the discussion of stratospheric dynamics is distracting the reader. Given the aim of the paper to present and understand the local mesospheric CO behavior, the detailed presentation of stratospheric circulation patterns in figures 3, 5 and 6 and related texts does not help this understanding and should be erased in favor of a concise discussion of joint stratosphere-mesosphere data in the sense of the comment above.

TECHNICAL CORRECTIONS

L19: Change "was" to "is" for present tense.

L23: Change "have been" to "are".

L42: Change "happened" to "happens".

L45: Change "The" to "A".

L51: Insert "is" after "database".

L56: Replace "waves" by "activity" and "propagate" by "propagates"

L58: What do you mean with "upward transfer of the momentum and heat"? The EP flux is the flux of wave activity, which is much upward and equatorward in the case of SSWs. Please, reformulate.

L61: Insert "its" after "is".

L65: Change "atmosphere" to "atmospheric".

L84: Replace "in" with "associated with", for example.

L143: Replace "are" with "is" because it refers to "data set" (singular).

L156: Insert a comma after "snow".

L160: Why not use "CO" for "carbon dioxide" (here and elswhere)?

L161: Are these validation tests published and documented? Please, provide a reference.

L162: Why not use "MWR" for "microwave radiometer" (here and elswhere)?

L166: Above you wrote "sideband" - please, unify.

L167: Replace "The first" with "At first".

L169: Replace "The second" with "At second".

L175: Which of the two methods were used for this article?

L178: What do you mean with "similar"? Rüfenachts instrument is for ozone at 30-79 km - please, specify yours.

L185: Insert "and" before "have".

L200: Shorten subtitle to "3 Northern hemisphere SSW effects" in the same style as for section 4.

L202: Begin with "The general" and insert "a" after "is".

L204: Insert "a" before "sequence".

L212: Why not use "SSW" for "sudden stratospheric warming" (here and elswhere)?

L225: Replace "in the" with "into".

L233: See comment for L58.

L238: Replace "since" with "after".

L242: "QBO" appears only once in the text, so it doesn not deed to be abbreviated.

L244: Insert "less likely" after "latitudes".

L256: Correct "WMR" to "VMR".

L376: Insert "The" before "Elevation".

L377: Insert "A" before "Similar".

L437: The paragraph should be shifted to where first reference to Fig. 4 is made.

Fig.2: Please, mark "10 Feb" in (a) as in (b),(c), and (d).

Fig. 4: Please, indicate height ranges in (a) and (b) as in (c) and (d).

Fig. 7: Please, indicate "10 Feb" as in Fig. 2.
* * *

---

## Referee Comment (RC2) · Anonymous Referee #2 · 16 Feb 2019

Comments on "Winter 2018 major sudden stratospheric warming impact on midlatitude mesosphere from microwave radiometer measurements" by Yuke Wang, Valery Shulga, Gennadi Milinevsky, et al. (2019).

General comment:

This study used the reanalysis and microwave radiometer measurements to investigate the February 2018 SSW event and its impact on the midlatitude mesosphere from an in-situ site. Given that the application of radiometers measurements into the SSW studies is still lack, this manuscript is suitable for the Atmospheric Chemistry and Physics journal. However, I have a concern about the novelty of the paper. The SSW events,

their impacts, and predictions have been widely explored in literature, but this study shows little review on the previous studies (e.g., Charlton et al. 2007JC; Hu, Ren, et al. 2014JAS; Taguchi 2018JGR; Rao et al. 2018JGR; Tripathi et al. 2016MWR; Karpechko et al. 2018GRL; Rao et al. 2019AAS...). By comparing the previous studies and this one, the novelty of this study can be well stressed in the introduction and discussion sections. The authors are responsible for their investigating the latest publications about this topic on their own. In addition, some typos and description errors still exist in the manuscript. The structure of this version can be further improved. Therefore, I recommend a major-plus revision. If those problems are well solved, the ACP journal can consider its publication. Please see my specific comments below.

Major comments:

1. The English language needs to be further improved. Many weird expressions can be found in this manuscript. I will list all of them in the minor comments one by one. I found some English speakers in the coauthors. Send this manuscript to all of them and well polish the language and correct all typos.

2. The organization of the manuscript is disappointing. I found many data links in the main body of the "paper" (Lines 236, 243, 415, 741, 743, 748, 779 ......). Why not introduce all the datasets in the method section? Or present a table to list all the data sources the authors used. The random placement of any dataset largely diminishes the general quality of the manuscript. The authors are writing a scientific article, not a diary.

3. The discussion section seems to be a replication of the former sections, not a real discussion at all. The authors need to compare their result with others (e.g., Taguchi 2018; Karpechko et al. 2018; Rao et al. 2019......) to put an emphasis on the new finding of the study. Another concern is about the discussion on the stratopause elevation, descent, and disappearance, which are not shown in any figure. If the stratospause is drawn in relevant figures (Figs. 2, 4 and 6), their description will be more easily

understood.

4. Some low-level mistakes should be avoided as much as possible. For example, it is well known that the ERA Interim has 37 levels from 1000 hPa to 1 hPa (Dee et al, 2011), but the authors can extend this reanalysis to 0.01 hPa (Line 188) and 0.1 hPa (Line 196). They did not refer to ERA Interim website for accurate introduction, but did to the second-hand data source (Line 197). Where is the 60-layer ERA Interim (Line 196) dataset from? Similar problems can also be found in other data sources. Please carefully check, check and recheck, and search, search and re-search. The authors should make sure that they present a real research.

Minor comments:

1. L22: ERA Interim, NCEP/NCAR, and other second-hand datasets that have been processed by others. Right?

2. L23: reanalyzes => reanalyses.

3. L42: Awkward expression for "the event of SSW" => the SSW event.

4. L46, L49: Weird phrase for "eastward", "westward"=> westerlies, easterlies.

5. L51: The citations are not exhaustive. Other studies should be well reviewed.

6. L52-55: Weird transition for the topics. What is the "usefull tool"? Can you provide more information for readers?

7. L58-59: The explanation is wrong. The upward propagations of waves do not denote the upward propagation of momentum and heat. The EP flux components are nearly proportional to the eddy momentum flux and eddy heat flux, so the poleward eddy heat flux favors the polar warming. It is wrong to stress the upward transfer of heat.

8. L61: ". . . is impact . . ." => is its impact

9. L63: "during the weeks or even month" =>in weeks or even more than one month

10. L72: upward displacements or downward displacements?

11. L89-90: Are you sure de la Torre et al. (2012) developed the WACCM and Shepherd at al. (2014) developed the CMAM. If they are the model users, please rephrase.

12. L92: for example=> e.g.

13. L99: exchange in stratosphere-mesosphere coupling => exchange between the stratosphere and the mesosphere?

14. L101: the => a

15. L111-112: Use the past tense.

16. L127: change "sub-vortices" to "sister vortices."

17. L128: Few articles use "eastward" and "westward" to denote the zonal wind direction. The wind direction refers to where the wind comes from, not where the wind will go. Did the authors learn some lessons like an introduction to meteorology?

18. L130-L132: It is not true. Rao et al. (2018) first reported the February 2018 SSW, followed by Karpechko et al. (2018), right?

19. L133: Use the present tense.

20. L143: in the 2017/18 winter?

21: L144: Weird expression. Consider? Please rephrase.

22. L145: What does "this" refer to?

23. L152: You think midlatitudes are in Kharkiv? It will be better to change to "Kharkiv in midlatitudes".

24. L159: Please check the ACP citation format. "Piddyachiy et al. 2010; Piddyachiy et al. 2017"=> Piddyachiy et al. 2010, 2017

25. L160: Delete "observations".

26. L162: Wrong citation format.

27. L167, 169: The first, the second=> Firstly, secondly.

28. L168: by search . . .=> by searching

29. L178: similar to=> consistent with

30. L188, L196: 0.01hPa? 0.1hPa? 60 layers?

31. L190-191: Rephrase this sentence.

32. L202: What do you mean by "general tendency"?

33. L205: in the 2017/18 winter.

34. L207: Why is it "enhanced warming"? It will mislead readers that the warming is enhanced. Similar problems can be also seen in Line 211.

35. L212: Use the present tense. Reverse=> reversal

36. L219-220, 226: Rao et al. (2018) also studied the January 2009 SSW.

37. L225-226: You can just say: "the SSW in February 2018 is mainly forced by wave 2"

38. L227: The description is not clear. Maybe you can describe something like this: The domination of planetary waves changes from wave 1 in January to wave 2 in February.

39. L230-231: This conclusion contradicts with the 2018 February SSW type. The vortex split SSW events are mainly caused by the wave 2. If you diagnose the poleward eddy heat flux at 100 hPa, you will not reach such conclusion.

40. L232: Delete "vector".

41. L233-234: This expression is wrong. The EP flux is parallel to the planetary wave propagation, but cannot measure the upward transfer of heat flux and momentum flux.

The eddy heat flux ([T'v']) and eddy momentum fluxes ([u'v']) are a good measure of meridional transport, NOT VERTICAL transport.

42. L235: The EP flux is a second-hand product based on NCEP/NCAR reanalysis and redistributed by ESRL. The data should be introduced in the method section. If the authors calculate the EP flux themselves, the procedure should also be described.

43. L241: I draw the QBO evolution based on NCEP/NCAR reanalysis and do not find the same conclusion as the authors obtained. Moreover, if the QBO is in its easterly phase, the equatorward transport of waves is suppressed (Holton and Tan 1980). The authors are responsible for their correct explanation of their results.

44. L260: What does the 6-ppmb level mean? I guess the authors mean the 6-ppmb contour. Moreover, the figure caption does not depict the white contour. Add the caption for the white contours and delete "(thin parts . . . . . . )"

45. L265: Change the citation format. It will be better to revise to "[Fig. 4 in Koo et al. (2017); Fig. 5 in Rayan et al. (2017)]

46. L269: I can not understand why the authors used weird phase like "eastward direction". What if you just use "westerly winds"?

47. L278: Have you compared the effects of the February 2018 SSW event in different regions? If not, how can you get such conclusion?

48. L284: What is "this Z"? Do you have "that Z"? Revised to "The Z . . ."

49. L285: What do you mean by "that"?

50. L291-294: This long sentence is not easy to understand. Please split and rephrase.

51. L299-300: Did you perform a power spectra analysis? The authors metioned the "5-8 days period".

53. L302: What are the atmospheric normal modes?

[Figure]

54. L308: the easterly winds.

55. L313: What are the "reverse process"?

56. L316: What are the "meridional tendency"?

57. L328: I found that the authors did not know where they should put the adverb "also". Please shift it after "are". Revise throughout the manuscript.

58. L334: Relative to what the influence weakens?

59. Section 5: This section is result indeed, not a real discussion. Please add more discussion on the differences between this study and previous studies (Rao et al. 2018JGR; Karpechko et al. 2018GRL . . . . . .). Emphasize the novelty of your work.

60. L341: westward . . . wind => mesospheric easterly winds

61. L342: Only wind speed? I don't think so.

62. L345-346: Replicate the method. Is there any new information?

63. L348: Reanalyzes (verb) => reanalyses (noun)

64. L355, L456: The stratopause variation was not explored at all. How did the authors come to the conclusion on "stratopause disappearance"?

65. L357: Ambiguous phrase "increase in winter season".

66. L369: What is "the similar processes"? Rather ambiguous.

67. L372: "This Z" => "The . . ."

68. L378: Who noted this? Could you add a citation?

69. L387: Delete "meridian".

70. L389: Did you show this?

71. L400: Did you mean the blocking high?

72. L404-406: Rephrase this sentence.

73. L407: again? How many replacements of warming by cooling happened?

74. L414: What are the large-scale process? The authors seemed to hide some useful information. Another data link appears. The organization of the manuscript needs to be improved.

75. L419: What do the previous studies refer to? Add the citations directly.

76. L435: Add a comma after "(2012)" and after "(2014)".

77. L450: registered? Revised to "documented".

78. L466: Add "is" after "which".

References: Rao, J., R.-C. Ren, H. Chen, X. Liu, Y. Yu, and Y. Yang, 2019: Subseasonal to seasonal hindcasts of stratospheric sudden warming by BCC_CSM1.1(m): A comparison with ECMWF. Adv. Atmos. Sci., accepted. doi: 10.1007/s00376-018-8165-8. Charlton, A. J., and L. M. Polvani, 2007: A new look at stratospheric sudden warmings. Part I: Climatology and modeling benchmarks. Journal of Climate, 20, 449-469. Hu, J., Ren, R., & Xu, H. (2014). Occurrence of winter stratospheric sudden warming events and the seasonal timing of spring stratospheric final warming. Journal of the Atmospheric Sciences, 71(7), 2319-2334, doi:10.1175/JAS-D-13-0349.1. Rao, J., R. Ren, H. Chen, Y. Yu, and Y. Zhou, 2018: The Stratospheric Sudden Warming Event in February 2018 and its Prediction by a Climate System Model. J. Geophys. Res. Atmos., 123(23), 13332-13345, doi: 10.1029/2018JD028908. Taguchi, M., 2018: Comparison of Subseasonal-to-Seasonal Model Forecasts for Major Stratospheric Sudden Warmings. Journal of Geophysical Research: Atmospheres, 123, 10,231-210,247, doi:10.1029/2018jd028755. Karpechko, A. Y., A. Charlton‐Perez, M. Balmaseda, N. Tyrrell, and F. Vitart, 2018: Predicting Sudden Stratospheric Warming 2018 and its Climate Impacts with a Multi‐Model Ensemble. Geophysical Research Letters, 45, 13,538-513,546, doi:10.1029/2018gl081091. Tripathi, O. P., and

Coauthors, 2016: Examining the Predictability of the Stratospheric Sudden Warming of January 2013 Using Multiple NWP Systems. Monthly Weather Review, 144, 1935-1960, doi:10.1175/mwr-d-15-0010.1.
* * *

---

## Author Comment (AC1) · 18 May 2019

Reply to Referee #1

Ref: Atmos. Chem. Phys. Discuss., https://doi.org/10.5194/acp-2018-1361-RC1, Apr 04, 2019 Title: "Winter 2018 major sudden stratospheric warming impact on midlatitude mesosphere from microwave radiometer measurements" by Yuke Wang et al.

We thank Referee #1 for useful comments, discussion and proposals for corrections. We made corrections and changed the text according to the suggestions. Our revisions and reply to Referee #1 comment are below. Line numbers of the revised manuscript are indicated.

Referee Comment: RC Authors Comment: AC

RC: GENERAL COMMENTS

The topic of ground-based mesospheric carbon dioxide observations is interesting and fits into the scope of ACP. The local microwave radiometer observations over Kharkiv during the major warming event in 2018 are valuable, but they need to be validated and explained adequately. In this sense I suggest the authors to add stratospheric carbon dioxide and mesospheric temperature data for a solid discussion of horizontal and vertical transports. If this information is provided in a concentrated form during a *major revision*, the paper would gain further value.

AC: CO and mesospheric temperature data horizontal and vertical transports are included in new Sect. 4.1 and 4.3: 4.1 CO variability 4.3 Temperature changes.

RC: SPECIFIC COMMENTS

RC: 1) Validation: A validation of the reported CO observations is required. For this purpose satellite observations could be used 1) to compare the CO in the 60-90 km range and 2) to extend the profiles will into the stratosphere. This would allow the qualification of downward transport during the 2018 major warming event in comparison with other events. Note, that the shown changes remain in the 70-80 km range while Funke et al. (2009) reports on effects down to 30 km for the 2004 major warming event. Such a three-dimensional picture would allow to place the local observations into a global context.

AC: Comparison with MLS data is now included (Figure 3). This shows good agreement. We extend CO profiles into the stratosphere (see new Figure 3).

RC: 2) Explanation: The present explanation of the observed mesospheric is highly speculative. It uses analysis data for the stratosphere which does not directly imply a clear picture of the mesosphere. So you can not use the presence of stratospheric planetary waves to explain oscillations in the mesosphere on the daily basis. For such a link mesospheric data are required, which exist with satellites.

AC: Satellite mesospheric data from MLS is now included for explanation in a new text in Sect. 4.1, 4.3, 4.4 and Supplement.

In this sense, you give some information on MLS-derived temperatures in figures 2.a and 7, but the latter is very noisy and not very helpful. The presentation of maps at selected levels would considerably help the interpretation of local CO and T observations in terms of three-dimensional transports.

AC: Maps of the CO distribution at selected level have been created and they are described in new Sect. 4.1 (Figure 4, Figure S1 and Figure S2). Along with CO profiles (Figure 3) they are used for the CO motion analysis in terms of horizontal and vertical transports.

RC: 3) Concentration: In some points the discussion of stratospheric dynamics is distracting the reader. Given the aim of the paper to present and understand the local mesospheric CO behavior, the detailed presentation of stratospheric circulation patterns in figures 3, 5 and 6 and related texts does not help this understanding and should be erased in favor of a concise discussion of joint stratosphere-mesosphere data in the sense of the comment above.

AC: Figures 3, 5 and 6, and the discussion of stratospheric dynamics has been erased.

RC: TECHNICAL CORRECTIONS

L19: Change "was" to "is" for present tense. AC: Corrected L23: Change "have been" to "are". AC: Corrected L42: Change "happened" to "happens". AC: Corrected L45: Change "The" to "A". AC: Corrected L51: Insert "is" after "database". AC: Corrected in L53. L56: Replace "waves" by "activity" and "propagate" by "propagates" AC: Corrected in L60. L58: What do you mean with "upward transfer of the momentum and heat"? The EP flux is the flux of wave activity, which is much upward and equatorward in the case of SSWs. Please, reformulate. AC: Old Figure 2 and corresponding text have been removed. L61: Insert "its" after "is". AC: Corrected in L64. L65: Change "atmosphere" to "atmospheric". AC: Corrected in L68. L84: Replace "in" with "associated with", for example. AC: Replaced in L94. L143: Replace "are" with "is" because it refers to "data set" (singular). AC: Corrected in L152. L156: Insert a comma after "snow". AC: Inserted in L166. L160: Why not use "CO" for "carbon dioxide" (here and elsewhere)? AC: Corrected in L169 and hereinafter. L161: Are these validation tests published and documented? Please, provide a reference. AC: Reference provided in L170. L162: Why not use "MWR" for "microwave radiometer" (here and elsewhere)? AC: Corrected in L174 and hereinafter. L166: Above you wrote "sideband" - please, unify. AC: Corrected in L174. L167: Replace "The first" with "At first". AC: L175: Corrected to "Firstly" as proposed by Referee #2. L169: Replace "The second" with "At second". AC: L177: Corrected to "Secondly" as proposed by Referee #2. L175: Which of the two methods were used for this article? AC: Explained in L182–184. In this article, the both methods were used with averaging the values of the zonal wind speed for altitudes of 70–85 km. Time interval January 1 – March 31, 2018 including the SSW 2018 event is considered. L178: What do you mean with "similar"? Rüfenachts instrument is for ozone at 30-79 km - please, specify yours. AC: This paragraph has been removed. L185: Insert "and" before "have". AC: Inserted in L191. L200: Shorten subtitle to "3 Northern hemisphere SSW effects" in the same style as for section 4. AC: Shortened, L210. L202: Begin with "The general" and insert "a" after "is". AC: Sentence rephrased in L212. L204: Insert "a" before "sequence". AC: Inserted and rephrased in L214. L212: Why not use "SSW" for "sudden stratospheric warming" (here and elswhere)? AC: We use “SSW” in L227 and below. L225: Replace "in the" with "into". AC: Sentence rephrased in L229–230. L233: See comment for L58. AC: The text corrected. The EP-flux discussion removed. L238: Replace "since" with "after". AC: The text removed. L242: "QBO" appears only once in the text, so it does not indeed to be abbreviated. AC: The text removed. L244: Insert "less likely" after "latitudes". AC: The text removed. L256: Correct "WMR" to "VMR". AC: Corrected in

L256. L376: Insert "The" before "Elevation". AC: This paragraph has been removed. L377: Insert "A" before "Similar". AC: This paragraph has been removed. L437: The paragraph should be shifted to where first reference to Fig. 4 is made. AC: This paragraph has been removed. Fig.2: Please, mark "10 Feb" in (a) as in (b),(c), and (d). AC: Corrected in all Figures. Fig. 4: Please, indicate height ranges in (a) and (b) as in (c) and (d). AC: Corrected, indicated in new Figure 3. Fig. 7: Please, indicate "10 Feb" as in Fig. 2. AC: Corrected in all Figures.

Reply to Referee #2

Ref: Atmos. Chem. Phys. Discuss., https://doi.org/10.5194/acp-2018-1361-RC2, Mar 16, 2019 Title: “Winter 2018 major sudden stratospheric warming impact on midlatitude mesosphere from microwave radiometer measurements” by Yuke Wang et al.

We thank Referee #2 for useful comments, discussion and proposals for corrections. We made corrections and changed the text according suggestions. Our revisions and reply to Referee #2 comments are below. RC (AC) is Referee (Authors) comments. Line numbers of the revised manuscript are indicated.

Referee #2 Comments: General comment:

RC: This study used the reanalysis and microwave radiometer measurements to investigate the February 2018 SSW event and its impact on the midlatitude mesosphere from an in-situ site. Given that the application of radiometers measurements into the SSW studies is still lack, this manuscript is suitable for the Atmospheric Chemistry and Physics journal. However, I have a concern about the novelty of the paper. The SSW events, their impacts, and predictions have been widely explored in literature, but this study shows little review on the previous studies (e.g., Charlton et al. 2007JC; Hu, Ren, et al. 2014JAS; Taguchi 2018JGR; Rao et al. 2018JGR; Tripathi et al. 2016MWR; Karpechko et al. 2018GRL; Rao et al. 2019AAS). By comparing the previous studies and this one, the novelty of this study can be well stressed in the introduction and discussion sections. The authors are responsible for their investigating the latest publications about this topic on their own. In addition, some typos and description errors still exist in the manuscript. The structure of this version can be further improved. Therefore, I recommend a major-plus revision. If those problems are well solved, the ACP journal can consider its publication. Please see my specific comments below.

AC: We (i) added new results in the revised manuscript (Fig. 3, Fig. 4, Figs. 6–9 and Supplemental Material), (ii) better structured Sections of the results and discussion, (iii) highlighted the novelty and (iv) corrected typos and errors.

Major comments: RC1. The English language needs to be further improved. Many weird expressions can be found in this manuscript. I will list all of them in the minor comments one by one. I found some English speakers in the coauthors. Send this manuscript to all of them and well polish the language and correct all typos.

AC1: We have improved the grammar in the manuscript and corrected errors.

RC2. The organization of the manuscript is disappointing. I found many data links in the main body of the "paper" (Lines 236, 243, 415, 741, 743, 748, 779 : : :: : :). Why not introduce all the datasets in the method section? Or present a table to list all the data sources the authors used. The random placement of any dataset largely diminishes the general quality of the manuscript. The authors are writing a scientific article, not a diary.

AC2: We have moved description of the data sets to sub-Section 2.2 and, partially, to Supplement.

RC3. The discussion section seems to be a replication of the former sections, not a real discussion at all. The authors need to compare their result with others (e.g., Taguchi 2018; Karpechko et al. 2018; Rao et al. 2019: : :: : :) to put an emphasis on the new finding of the study. Another concern is about the discussion on the stratopause elevation, descent, and disappearance, which are not shown in any figure. If the stratospause is drawn in relevant figures (Figs. 2, 4 and 6), their description will be more easily understood.

AC3: We have improved the structure of the discussion (new sub-Sections 5.1–5.3 have been introduced) and conclusions as suggested.

RC4. Some low-level mistakes should be avoided as much as possible. For example, it is well known that the ERA Interim has 37 levels from 1000 hPa to 1 hPa (Dee et al, 2011), but the authors can extend this reanalysis to 0.01 hPa (Line 188) and 0.1 hPa (Line 196). They did not refer to ERA Interim website for accurate introduction, but did to the second-hand data source (Line 197). Where is the 60-layer ERA Interim (Line 196) dataset from? Similar problems can also be found in other data sources. Please carefully check, check and recheck, and search, search and re-search. The authors should make sure that they present a real research.

AC4: We have corrected the specific errors and omissions noted above, and made other improvements for consistency. Details of the pressure level ranges are described in Supplemental Section "The description of the data used for analysis".

Minor comments: RC1. L22: ERA Interim, NCEP/NCAR, and other second-hand datasets that have been processed by others. Right? AC1: L22 corrected to "Data from the ERA-Interim and MERRA-2 reanalyses,. . .". Other data sources are described in Section 2.2. RC2. L23: reanalyzes => reanalyses. AC2: Corrected in L22. RC3. L42: Awkward expression for "the event of SSW" => the SSW event. AC3: L42 Corrected to "Major sudden stratospheric warming (SSW) events which happen roughly. . ." RC4. L46, L49: Weird phrase for "eastward", "westward"=> westerlies, easterlies. AC4: Corrected throughout the manuscript. RC5. L51: The citations are not exhaustive. Other studies should be well reviewed. AC5: Other studies cited in L47–53. RC6. L52-55: Weird transition for the topics. What is the "usefull tool"? Can you provide more information for readers? AC6: More information is provided in L57–59. RC7. L58-59: The explanation is wrong. The upward propagations of waves do not denote the upward propagation of momentum and heat. The EP flux components are nearly proportional to the eddy momentum flux and eddy heat flux, so the poleward eddy heat flux favors the polar warming. It is wrong to stress the upward transfer of heat. AC7: L62–63, the text corrected to “The enhanced wave activity results in the rapid warming of the polar stratosphere and the breakdown of the stratospheric polar vortex. . .” RC8. L61: ": : : is impact : : :" => is its impact . AC: Corrected in L64. RC9. L63: "during the weeks or even month" =>in weeks or even more than one month AC9: L66, corrected to “during the following weeks to month”. RC10. L72: upward displacements or downward displacements? AC10: Detailed in L75–80. RC11. L89-90: Are you sure de la Torre et al. (2012) developed the WACCM and Shepherd at al. (2014) developed the CMAM. If they are the model users, please rephrase. AC11: L98–100, The text rewritten. RC12. L92: for example=> e.g. AC12: Corrected in L102. RC13. L99: exchange in stratosphere-mesosphere coupling => exchange between the stratosphere and the mesosphere? AC13: Corrected in L108–109. RC14. L101: the => a AC14: Corrected in L110. RC15. L111-112: Use the past tense. AC15: Corrected in L119–120. RC16. L127: change "sub-vortices" to "sister vortices." AC16: Changed in L136. RC17. L128: Few articles use "eastward" and "westward" to denote the zonal wind direction. The wind direction refers to where the wind comes from, not where the wind will go. Did the authors learn some lessons like an introduction to meteorology? AC17: Corrected to “westerly” and “easterly” throughout the text. RC18. L130-L132: It is not true. Rao et al. (2018) first reported the February 2018 SSW, followed by Karpechko et al. (2018), right? AC18: Corrected to (Rao et al., 2018; Karpechko et al., 2018; Vargin and Kiryushov, 2019) in L138. RC19. L133: Use the present tense. AC19: Corrected, L143. RC20. L143: in the 2017/18 winter? AC20: Corrected, L153. RC21: L144: Weird expression. Consider? Please rephrase. AC21: Rephrased, L152–153. RC22. L145: What does "this" refer to? AC22: Removed, L153. RC23. L152: You think midlatitudes are in Kharkiv? It will be better to change to "Kharkiv in midlatitudes". AC23: Changed, L162. RC24. L159: Please check the ACP citation format. "Piddyachiy et al. 2010; Piddyachiy et al. 2017"=> Piddyachiy et al. 2010, 2017 AC24: Corrected, L168. RC25. L160: Delete "observations". AC25: Rewritten, L169. RC26. L162: Wrong citation format. AC26: Corrected, L170. RC27. L167, 169: The first, the second=> Firstly, secondly. AC27: Corrected, L175, L177. RC28. L168: by search : : :=> by searching AC28: Corrected, L176. RC29. L178: similar to=> consistent with AC29: Rewritten, 182–184. RC30. L188, L196: 0.01hPa? 0.1hPa? 60 layers? AC30: Detailed in Supplement, page 1. RC31. L190-191: Rephrase this sentence. AC31: Changed in revised Section 2.2. RC32. L202: What do you mean by "general tendency"? AC32: L 212–214, corrected to: “Descending air masses are observed throughout the mesosphere and stratosphere of the winter polar region (Orsolini et al., 2010; Chandran and Collins, 2014; Limpasuvan et al., 2016; Zülicke et al., 2018).” RC33. L205: in the 2017/18 winter. AC33: Corrected, L215. RC34. L207: Why is it "enhanced warming"? It will mislead readers that the warming is enhanced. Similar problems can be also seen in Line 211. AC34: Removed in both cases, L215–217. RC35. L212: Use the present tense. Reverse=> reversal AC35: Corrected, L229. RC36. L219-220, 226: Rao et al. (2018) also studied the January 2009 SSW. AC36: Corrected, L236. RC37. L225-226: You can just say: "the SSW in February 2018 is mainly forced by wave 2" AC37: Rewritten, L229–230. RC38. L227: The description is not clear. Maybe you can describe something like this: The domination of planetary waves changes from wave 1 in January to wave 2 in February. AC38: Description improved L232–234. RC39. L230-231: This conclusion contradicts with the 2018 February SSW type. The vortex split SSW events are mainly caused by the wave 2. If you diagnose the poleward eddy heat flux at 100 hPa, you will not reach such conclusion. AC39: This fragment has been removed. RC40. L232: Delete "vector". AC40: This paragraph has been removed. RC41. L233-234: This expression is wrong. The EP flux is parallel to the planetary wave propagation, but cannot measure the upward transfer of heat flux and momentum flux. The eddy heat flux ([T’v’]) and eddy momentum fluxes ([u’v’]) are a good measure of meridional transport, NOT VERTICAL transport. AC41: This paragraph has been removed. The detailed study of EP flux behavior during the event left for future consideration. RC42. L235: The EP flux is a second-hand product based on NCEP/NCAR reanalysis and redistributed by ESRL. The data should be introduced in the method section. If the authors calculate the EP flux themselves, the procedure should also be described. AC42: This paragraph has been removed. RC43. L241: I draw the QBO evolution based on NCEP/NCAR reanalysis and do not find the same conclusion as the authors obtained. Moreover, if the QBO is in its easterly phase, the equatorward transport of waves is suppressed (Holton and Tan 1980). The authors are responsible for their correct explanation of their results. AC43: The results related to EP-flux have been removed. RC44. L260: What does the 6-ppmb level mean? I guess the authors mean the 6-ppmb contour. Moreover, the figure caption does not depict the white contour. Add the caption for the white contours and delete "(thin parts : : :: : : )" AC44: Corrected, L260, L263. Figure 3 caption updated, "thin parts..." deleted. RC45. L265: Change the citation format. It will be better to revise to "[Fig. 4 in Koo et al. (2017); Fig. 5 in Rayan et al. (2017)] AC45: Corrected, L267–268. RC46. L269: I can not understand why the authors used weird phase like "eastward direction". What if you just use "westerly winds"? AC46: Corrected to “westerly wind” throughout the manuscript. RC47. L278: Have you compared the effects of the February 2018 SSW event in different regions? If not, how can you get such conclusion? AC47: Corrected, L253–255, and Fig. 4 is introduced for illustration. RC48. L284: What is "this Z"? Do you have "that Z"? Revised to "The Z : : :" AC48: This fragment has been removed. RC49. L285: What do you mean by "that"? AC49: This paragraph has been deleted. RC50. L291-294: This long sentence is not easy to understand. Please split and rephrase. AC50: This paragraph has been deleted RC51. L299-300: Did you perform a power spectra analysis? The authors mentioned the "5-8 days period". AC51: Spectral analysis provided and results included in new Section 4.4. RC53. L302: What are the atmospheric normal modes? AC53: This paragraph has been removed. RC54. L308: the easterly winds. AC54: This paragraph has been removed. RC55. L313: What are the "reverse process"? AC55: This paragraph has been removed. RC56. L316: What are the "meridional tendency"? AC56: This paragraph has been removed. RC57. L328: I found that the authors did not know where they should put the adverb "also". Please shift it after "are". Revise throughout the manuscript. AC57: Corrected.

RC58. L334: Relative to what the influence weakens? AC58: Old Fig. 6 and related text have been removed. RC59. Section 5: This section is result indeed, not a real discussion. Please add more discussion on the differences between this study and previous studies (Rao et al. 2018JGR; Karpechko et al. 2018GRL : : :: : :). Emphasize the novelty of your work. AC59. Discussion has been updated; see new Section 5 and sub-Sections 5.1–5.3. RC60. L341: westward : : : wind => mesospheric easterly winds AC60: Corrected and rewritten in L433, see AC61. RC61. L342: Only wind speed? I don't think so. AC61: The CO altitude profiles in the mesosphere have been measured by the MWR with one day time resolution. Using the CO molecule as tracer, the wind speed has been retrieved from the Doppler shift of the CO 115.3 GHz emission and the mesospheric winds reverse from westerly to easterly below the winter mesopause region (70–85 km) has been detected. L431-434 RC62. L345-346: Replicate the method. Is there any new information? AC62: Sentence rephrased in L431–434. RC63. L348: Reanalyzes (verb) => reanalyses (noun) AC63: Corrected in L438. RC64. L355, L456: The stratopause variation was not explored at all. How did the authors come to the conclusion on "stratopause disappearance"? AC64: The stratopause variations are described in new Section 4.3. RC65. L357: Ambiguous phrase "increase in winter season". AC65: Deleted. RC66. L369: What is "the similar processes"? Rather ambiguous. AC66: Removed. RC67. L372: "This Z" => "The : : :" AC67: Removed. RC68. L378: Who noted this? Could you add a citation? AC68: Removed. RC69. L387: Delete "meridian". AC69: Deleted. The changes in the CO field are illustrated in new Fig. 4, Fig. S1 and S2 and are described in related text. RC70. L389: Did you show this? AC70: The results to confirm this are included in new Section 4.1 and are discussed in Section 5.1. RC71. L400: Did you mean the blocking high? AC71: Removed. Analysis is focused on the CO re-distribution at different pressure levels in the context of the wave effects (Sections 4.1 and 5.1). RC72. L404-406: Rephrase this sentence. AC72: Removed. RC73. L407: again? How many replacements of warming by cooling happened? AC73: Rephrased in L522–523. RC74. L414: What are the large-scale process? The authors seemed to hide some useful information. Another data link appears. The organization of the manuscript needs to be improved. AC74: Removed. Zonal wave spectra are presented and discussed in new Sections 4.4 and 5.3. RC75. L419: What do the previous studies refer to? Add the citations directly. AC75: Citation added in L572–575. RC76. L435: Add a comma after "(2012)" and after "(2014)". AC76: This paragraph has been rephrased in L501–509. RC77. L450: registered? Revised to "documented". AC77: Corrected, L603. RC78. L466: Add "is" after "which". AC78: Corrected in L635.

Proposed references added and discussed.

References: Rao, J., R.-C. Ren, H. Chen, X. Liu, Y. Yu, and Y. Yang, 2019: Subseasonal to seasonal hindcasts of stratospheric sudden warming by BCC_CSM1.1(m): A comparison with ECMWF. Adv. Atmos. Sci., accepted. doi: 10.1007/s00376-018-8165-8. Charlton, A. J., and L. M. Polvani, 2007: A new look at stratospheric sudden warmings. Part I: Climatology and modeling benchmarks. Journal of Climate, 20, 449-469. Hu, J., Ren, R., & Xu, H. (2014). Occurrence of winter stratospheric sudden warming events and the seasonal timing of spring stratospheric final warming. Journal of the Atmospheric Sciences, 71(7), 2319-2334, doi:10.1175/JAS-D-13-0349.1. Rao, J., R. Ren, H. Chen, Y. Yu, and Y. Zhou, 2018: The Stratospheric Sudden Warming Event in February 2018 and its Prediction by a Climate System Model. J. Geophys. Res. Atmos., 123(23), 13332-13345, doi: 10.1029/2018JD028908. Taguchi, M., 2018: Comparison of Subseasonal-to-Seasonal Model Forecasts for Major Stratospheric Sudden Warmings. Journal of Geophysical Research: Atmospheres, 123, 10,231-210,247, doi:10.1029/2018jd028755. Karpechko, A. Y., A. CharltonâĖŸARĖĞ Perez, M. Balmaseda, N. Tyrrell, and F. Vitart, 2018: Predicting Sudden Stratospheric Warming 2018 and its Climate Impacts with a MultiâĖŸARĖĞModel Ensemble. Geophysical Research Letters, 45, 13,538-513,546, doi:10.1029/2018gl081091. Tripathi, O. P., Baldwin, M., Charlton-Perez, A., Charron, M., Cheung, J. C. H., Eckermann, S. D., Gerber, E., Jackson, D.R., Kuroda, Yu., Lang, A., McLay, J., Mizuta, R., Reynolds, C., Roff, G., Sigmond, M., Son, S.-W., and Stockdale, T. 2016. Examining the Predictability of the Stratospheric Sudden Warming of January 2013 Using Multiple NWP Systems. Monthly Weather Review, 144, 1935-1960, doi:10.1175/mwr-d-15-0010.1.

================

Our reply to Referee #1 and Referee #2 comments, the new text of the manuscript in the blue colored text and new figures, and the Supplement to the manuscript are given in the supplement below.

Please also note the supplement to this comment:
https://www.atmos-chem-phys-discuss.net/acp-2018-1361/acp-2018-1361-AC1-supplement.pdf

**Supplement:**

**Reply to Referee #1**

Ref: Atmos. Chem. Phys. Discuss., https://doi.org/10.5194/acp-2018-1361-RC1, Apr 04, 2019
Title: "Winter 2018 major sudden stratospheric warming impact on midlatitude mesosphere from microwave radiometer measurements" by Yuke Wang et al.

Dear Referee #1

We thank Referee #1 for useful comments, discussion and proposals for corrections. We made corrections and changed the text according to the suggestions. Our revisions and reply to Referee #1 comment are below in blue colored text.

**Referee Comment: RC**
**Authors Comment: AC**

RC: GENERAL COMMENTS

The topic of ground-based mesospheric carbon dioxide observations is interesting and fits into the scope of ACP. The local microwave radiometer observations over Kharkiv during the major warming event in 2018 are valuable, but they need to be validated and explained adequately. In this sense I suggest the authors to add stratospheric carbon dioxide and mesospheric temperature data for a solid discussion of horizontal and vertical transports. If this information is provided in a concentrated form during a \*major revision\*, the paper would gain further value.

**AC: CO and mesospheric temperature data horizontal and vertical transports are included in new Sect. 4.1 and 4.3:  4.1 CO variability 4.3 Temperature changes.**

RC: SPECIFIC COMMENTS

RC: 1) Validation: A validation of the reported CO observations is required. For this purpose satellite observations could be used 1) to compare the CO in the 60-90 km range and 2) to extend the profiles will into the stratosphere. This would allow the qualification of **downward transport** during the 2018 major warming event in comparison with other events. Note, that the shown changes remain in the 70-80 km range while Funke et al. (2009) reports on effects down to 30 km for the 2004 major warming event. Such a three-dimensional picture would allow to place the local observations into a global
context.

**AC: Comparison with MLS data is now included (Figure 3). This shows good agreement. We extend CO profiles into the stratosphere (see new Figure 3).**

RC: 2) Explanation: The present explanation of the observed mesospheric is highly speculative. It uses analysis data for the stratosphere which does not directly imply a clear picture of the mesosphere. So you can not use the presence of stratospheric planetary waves to explain oscillations in the mesosphere on the daily basis. For such a link mesospheric data are required, which exist with satellites**.**

**AC: Satellite mesospheric data from MLS is now included for explanation in a new text in Sect. 4.1, 4.3, 4.4 and Supplement**.

In this sense, you give some information on MLS-derived temperatures in figures 2.a and 7, but the latter is very noisy and not very helpful. The presentation of maps at selected levels would considerably help the interpretation of local CO and T observations in terms of three-dimensional transports.

**AC: Maps of the CO distribution at selected level have been created and they are described in new Sect. 4.1 (Figure 4, Figure S1 and Figure S2). Along with CO profiles (Figure 3) they are used for the CO motion analysis in terms of horizontal and vertical transports.**

RC: 3) Concentration: In some points the discussion of stratospheric dynamics is distracting the reader. Given the aim of the paper to present and understand the local mesospheric CO behavior, the detailed presentation of stratospheric circulation patterns in figures 3, 5 and 6 and related texts does not help this understanding and should be erased in favor of a concise discussion of joint stratosphere-mesosphere data in the sense of the comment above.

**AC: Figures 3, 5 and 6, and the discussion of stratospheric dynamics has been erased.**

RC: TECHNICAL CORRECTIONS

L19: Change "was" to "is" for present tense. Corrected

L23: Change "have been" to "are". Corrected

L42: Change "happened" to "happens". Corrected

L45: Change "The" to "A". Corrected

L51: Insert "is" after "database". Corrected in L53.

L56: Replace "waves" by "activity" and "propagate" by "propagates" Corrected in L60.

L58: What do you mean with "upward transfer of the momentum and heat"? The EP flux is the flux of wave activity, which is much upward and equatorward in the case of SSWs. Please, reformulate. AC: Old Figure 2 and corresponding text have been removed.

L61: Insert "its" after "is". Corrected in L64.

L65: Change "atmosphere" to "atmospheric". Corrected in L68.

L84: Replace "in" with "associated with", for example. Replaced in L94.

L143: Replace "are" with "is" because it refers to "data set" (singular). Corrected in L152.

L156: Insert a comma after "snow". Inserted in L166.

L160: Why not use "CO" for "carbon dioxide" (here and elsewhere)? Corrected in L169 and hereinafter.

L161: Are these validation tests published and documented? Please, provide a reference. Reference provided in L170.

L162: Why not use "MWR" for "microwave radiometer" (here and elsewhere)? Corrected in L174 and hereinafter.

L166: Above you wrote "sideband" - please, unify. Corrected in L174.

L167: Replace "The first" with "At first". L175: Corrected to "Firstly" as proposed by Referee #2.

L169: Replace "The second" with "At second". L177: Corrected to "Secondly" as proposed by Referee #2.

L175: Which of the two methods were used for this article?

AC: Explained in L182–184. In this article, the both methods were used with averaging the values of the zonal wind speed for altitudes of 70–85 km. Time interval January 1 – March 31, 2018 including the SSW 2018 event is considered.

L178: What do you mean with "similar"? Rüfenachts instrument is for ozone at 30-79 km - please, specify yours.

AC: This paragraph has been removed.

L185: Insert "and" before "have". Inserted in L191.

L200: Shorten subtitle to "3 Northern hemisphere SSW effects" in the same style as for section 4. Shortened, L210.
L202: Begin with "The general" and insert "a" after "is". Sentence rephrased in L212.
L204: Insert "a" before "sequence". Inserted and rephrased in L214.
L212: Why not use "SSW" for "sudden stratospheric warming" (here and elswhere)?
AC: We use "SSW" in L227 and below.
L225: Replace "in the" with "into". Sentence rephrased in L229–230.
L233: See comment for L58.
AC: The text corrected. The EP-flux discussion removed.
L238: Replace "since" with "after". The text removed.
L242: "QBO" appears only once in the text, so it does not indeed to be abbreviated. The text removed.
L244: Insert "less likely" after "latitudes". The text removed.
L256: Correct "WMR" to "VMR". Corrected in L256.
L376: Insert "The" before "Elevation". This paragraph has been removed.
L377: Insert "A" before "Similar". This paragraph has been removed.
L437: The paragraph should be shifted to where first reference to Fig. 4 is made. This paragraph has been
removed.
Fig.2: Please, mark "10 Feb" in (a) as in (b),(c), and (d). Corrected in all Figures.
Fig. 4: Please, indicate height ranges in (a) and (b) as in (c) and (d). Corrected, indicated in new Figure 3.
Fig. 7: Please, indicate "10 Feb" as in Fig. 2. Corrected in all Figures.
**Reply to Referee #2**
Ref: Atmos. Chem. Phys. Discuss., https://doi.org/10.5194/acp-2018-1361-RC2, Mar 16, 2019
Title: "Winter 2018 major sudden stratospheric warming impact on midlatitude mesosphere from microwave
radiometer measurements" by Yuke Wang et al.
We thank Referee #2 for useful comments, discussion and proposals for corrections. We made corrections and
changed the text according suggestions. Our revisions and reply to Referee #2 comments are below in blue
colored text. RC (AC) is Referee (Authors) comments. Line numbers of the revised manuscript are indicated.
**Referee #2 Comments:**
General comment:
**RC:** This study used the reanalysis and microwave radiometer measurements to investigate the February 2018
SSW event and its impact on the midlatitude mesosphere from an in-situ site. Given that the application of
radiometers measurements into the SSW studies is still lack, this manuscript is suitable for the Atmospheric
Chemistry and Physics journal. However, I have a concern about the novelty of the paper.
The SSW events, their impacts, and predictions have been widely explored in literature, but this study shows
little review on the previous studies (e.g., Charlton et al. 2007JC; Hu, Ren, et al. 2014JAS; Taguchi 2018JGR;
Rao et al. 2018JGR; Tripathi et al. 2016MWR; Karpechko et al. 2018GRL; Rao et al. 2019AAS). By comparing
the previous studies and this one, the novelty of this study can be well stressed in the introduction and
discussion sections. The authors are responsible for their investigating the latest publications about this topic on
their own. In addition, some typos and description errors still exist in the manuscript. The structure of this
version can be further improved. Therefore, I recommend a major-plus revision. If those problems are well
solved, the ACP journal can consider its publication. Please see my specific comments below.
**AC: We (i) added new results in the revised manuscript (Fig. 3, Fig. 4, Figs. 6–9 and Supplemental**
**Material), (ii) better structured Sections of the results and discussion, (iii) highlighted the novelty and (iv)**
**corrected typos and errors.**

Major comments:

**RC1.** The **English language** needs to be further improved. Many weird expressions can be found in this manuscript. I will list all of them in the minor comments one by one. I found some English speakers in the coauthors. Send this manuscript to all of them and well **polish the language** and correct all typos.

**AC1: We have improved the grammar in the manuscript and corrected errors.**

**RC2.** The organization of the manuscript is disappointing. I found many data links in the main body of the "paper" (Lines 236, 243, 415, 741, 743, 748, 779 : : :: : :). Why not introduce all the datasets in the method section? Or present a table to list all the data sources the authors used. The random placement of any dataset largely diminishes the general quality of the manuscript. The authors are writing a scientific article, not a diary.

**AC2: We have moved description of the data sets to sub-Section 2.2 and, partially, to Supplement.**

**RC3.** The discussion section seems to be a replication of the former sections, not a real discussion at all. The authors need to compare their result with others (e.g., Taguchi 2018; Karpechko et al. 2018; Rao et al. 2019: : :: : :) to put an emphasis on the new finding of the study. Another concern is about the discussion on the stratopause elevation, descent, and disappearance, which are not shown in any figure. If the stratospause is drawn in relevant figures (Figs. 2, 4 and 6), their description will be more easily understood.

**AC3: We have improved the structure of the discussion (new sub-Sections 5.1–5.3 have been introduced) and conclusions as suggested.**

**RC4.** Some low-level mistakes should be avoided as much as possible. For example, it is well known that the ERA Interim has 37 levels from 1000 hPa to 1 hPa (Dee et al, 2011), but the authors can extend this reanalysis to 0.01 hPa (Line 188) and 0.1 hPa (Line 196). They did not refer to ERA Interim website for accurate introduction, but did to the second-hand data source (Line 197). Where is the 60-layer ERA Interim (Line 196) dataset from? Similar problems can also be found in other data sources. Please carefully check, check and recheck, and search, search and re-search. The authors should make sure that they present a real research.

**AC4: We have corrected the specific errors and omissions noted above, and made other improvements for consistency. Details of the pressure level ranges are described in Supplemental Section "The description of the data used for analysis".**

Minor comments:

RC1. L22: ERA Interim, NCEP/NCAR, and other second-hand datasets that have been processed by others. Right?

AC1: L22 corrected to "Data from the ERA-Interim and MERRA-2 reanalyses,…". Other data sources are described in Section 2.2.

RC2. L23: reanalyzes => reanalyses.

AC2: Corrected in L22.

RC3. L42: Awkward expression for "the event of SSW" => the SSW event.

AC3: L42 Corrected to "Major sudden stratospheric warming (SSW) events which happen roughly…"

RC4. L46, L49: Weird phrase for "eastward", "westward"=> westerlies, easterlies.

AC4: Corrected throughout the manuscript.

RC5. L51: The citations are not exhaustive. Other studies should be well reviewed.

AC5: Other studies cited in L47–53.

RC6. L52-55: Weird transition for the topics. What is the "usefull tool"? Can you provide more information for readers?

AC6: More information is provided in L57–59.

RC7. L58-59: The explanation is wrong. The upward propagations of waves do not denote the upward propagation of momentum and heat. The EP flux components are nearly proportional to the eddy momentum flux and eddy heat flux, so the poleward eddy heat flux favors the polar warming. It is wrong to stress the upward transfer of heat.

AC7: L62–63, the text corrected to "The enhanced wave activity results in the rapid warming of the polar stratosphere and the breakdown of the stratospheric polar vortex…"

RC8. L61: ": : : is impact : : :" => is its impact .
AC: Corrected in L64.
RC9. L63: "during the weeks or even month" =>in weeks or even more than one month
AC9: L66, corrected to "during the following weeks to month".
RC10. L72: upward displacements or downward displacements?
AC10: Detailed in L75–80.
RC11. L89-90: Are you sure de la Torre et al. (2012) developed the WACCM and Shepherd at al. (2014)
developed the CMAM. If they are the model users, please rephrase.
AC11: L98–100, The text rewritten.
RC12. L92: for example=> e.g.
AC12: Corrected in L102.
RC13. L99: exchange in stratosphere-mesosphere coupling => exchange between the stratosphere and the
mesosphere?
AC13: Corrected in L108–109.
RC14. L101: the => a
AC14: Corrected in L110.
RC15. L111-112: Use the past tense.
AC15: Corrected in L119–120.
RC16. L127: change "sub-vortices" to "sister vortices."
AC16: Changed in L136.
RC17. L128: Few articles use "eastward" and "westward" to denote the zonal wind direction. The wind
direction refers to where the wind comes from, not where the wind will go. Did the authors learn some lessons
like an introduction to meteorology?
AC17: Corrected to "westerly" and "easterly" throughout the text.
RC18. L130-L132: It is not true. Rao et al. (2018) first reported the February 2018 SSW, followed by
Karpechko et al. (2018), right?
AC18: Corrected to (Rao et al., 2018; Karpechko et al., 2018; Vargin and Kiryushov, 2019) in L138.
RC19. L133: Use the present tense.
AC19: Corrected, L143.
RC20. L143: in the 2017/18 winter?
AC20: Corrected, L153.
RC21: L144: Weird expression. Consider? Please rephrase.
AC21: Rephrased, L152–153.
RC22. L145: What does "this" refer to?
AC22: Removed, L153.
RC23. L152: You think midlatitudes are in Kharkiv? It will be better to change to "Kharkiv in midlatitudes".
AC23: Changed, L162.
RC24. L159: Please check the ACP citation format. "Piddyachiy et al. 2010; Piddyachiy et al. 2017"=>
Piddyachiy et al. 2010, 2017
AC24: Corrected, L168.
RC25. L160: Delete "observations".
AC25: Rewritten, L169.
RC26. L162: Wrong citation format.
AC26: Corrected, L170.
RC27. L167, 169: The first, the second=> Firstly, secondly.
AC27: Corrected, L175, L177.
RC28. L168: by search : : :=> by searching
AC28: Corrected, L176.
RC29. L178: similar to=> consistent with
AC29: Rewritten, 182–184.
RC30. L188, L196: 0.01hPa? 0.1hPa? 60 layers?
AC30: Detailed in Supplement, page 1.
RC31. L190-191: Rephrase this sentence.
AC31: Changed in revised Section 2.2.
RC32. L202: What do you mean by "general tendency"?

AC32: L 212–214, corrected to: "Descending air masses are observed throughout the mesosphere and stratosphere of the winter polar region (Orsolini et al., 2010; Chandran and Collins, 2014; Limpasuvan et al., 2016; Zülicke et al., 2018)."

RC33. L205: in the 2017/18 winter.

AC33: Corrected, L215.

RC34. L207: Why is it "enhanced warming"? It will mislead readers that the warming is enhanced. Similar problems can be also seen in Line 211.

AC34: Removed in both cases, L215–217.

RC35. L212: Use the present tense. Reverse=> reversal

AC35: Corrected, L229.

RC36. L219-220, 226: Rao et al. (2018) also studied the January 2009 SSW.

AC36: Corrected, L236.

RC37. L225-226: You can just say: "the SSW in February 2018 is mainly forced by wave 2"

AC37: Rewritten, L229–230.

RC38. L227: The description is not clear. Maybe you can describe something like this: The domination of planetary waves changes from wave 1 in January to wave 2 in February.

AC38: Description improved L232–234.

RC39. L230-231: This conclusion contradicts with the 2018 February SSW type. The vortex split SSW events are mainly caused by the wave 2. If you diagnose the poleward eddy heat flux at 100 hPa, you will not reach such conclusion.

AC39: This fragment has been removed.

RC40. L232: Delete "vector".

AC40: This paragraph has been removed.

RC41. L233-234: This expression is wrong. The EP flux is parallel to the planetary wave propagation, but cannot measure the upward transfer of heat flux and momentum flux. The eddy heat flux ([T'v']) and eddy momentum fluxes ([u'v']) are a good measure of meridional transport, NOT VERTICAL transport.

AC41: This paragraph has been removed. The detailed study of EP flux behavior during the event left for future consideration.

RC42. L235: The EP flux is a second-hand product based on NCEP/NCAR reanalysis and redistributed by ESRL. The data should be introduced in the method section. If the authors calculate the EP flux themselves, the procedure should also be described.

AC42: This paragraph has been removed.

RC43. L241: I draw the QBO evolution based on NCEP/NCAR reanalysis and do not find the same conclusion as the authors obtained. Moreover, if the QBO is in its easterly phase, the equatorward transport of waves is suppressed (Holton and Tan 1980). The authors are responsible for their correct explanation of their results.

AC43: The results related to EP-flux have been removed.

RC44. L260: What does the 6-ppmb level mean? I guess the authors mean the 6-ppmb contour. Moreover, the figure caption does not depict the white contour. Add the caption for the white contours and delete "(thin parts : : :: : : )"

AC44: Corrected, L260, L263. Figure 3 caption updated, "thin parts..." deleted.

RC45. L265: Change the citation format. It will be better to revise to "[Fig. 4 in Koo et al. (2017); Fig. 5 in Rayan et al. (2017)]

AC45: Corrected, L267–268.

RC46. L269: I can not understand why the authors used weird phase like "eastward direction". What if you just use "westerly winds"?

AC46: Corrected to "westerly wind" throughout the manuscript.

RC47. L278: Have you compared the effects of the February 2018 SSW event in different regions? If not, how can you get such conclusion?

AC47: Corrected, L253–255, and Fig. 4 is introduced for illustration.

RC48. L284: What is "this Z"? Do you have "that Z"? Revised to "The Z : : :"

AC48: This fragment has been removed.

RC49. L285: What do you mean by "that"?

AC49: This paragraph has been deleted.

RC50. L291-294: This long sentence is not easy to understand. Please split and rephrase.

AC50: This paragraph has been deleted

RC51. L299-300: Did you perform a power spectra analysis? The authors mentioned the "5-8 days period".

AC51: Spectral analysis provided and results included in new Section 4.4.

RC53. L302: What are the atmospheric normal modes?
AC53: This paragraph has been removed.
RC54. L308: the easterly winds.
AC54: This paragraph has been removed.
RC55. L313: What are the "reverse process"?
AC55: This paragraph has been removed.
RC56. L316: What are the "meridional tendency"?
AC56: This paragraph has been removed.
RC57. L328: I found that the authors did not know where they should put the adverb "also". Please shift it after "are". Revise throughout the manuscript.
AC57: Corrected.
RC58. L334: Relative to what the influence weakens?
AC58: Old Fig. 6 and related text have been removed.
RC59. Section 5: This section is result indeed, not a real discussion. Please add more discussion on the differences between this study and previous studies (Rao et al. 2018JGR; Karpechko et al. 2018GRL : : :: : :). Emphasize the novelty of your work.
AC59. Discussion has been updated; see new Section 5 and sub-Sections 5.1–5.3.
RC60. L341: westward : : : wind => mesospheric easterly winds
AC60: Corrected and rewritten in L433, see AC61.
RC61. L342: Only wind speed? I don't think so.
AC61: The CO altitude profiles in the mesosphere have been measured by the MWR with one day time resolution. Using the CO molecule as tracer, the wind speed has been retrieved from the Doppler shift of the CO 115.3 GHz emission and the mesospheric winds reverse from westerly to easterly below the winter mesopause region (70–85 km) has been detected. L431-434
RC62. L345-346: Replicate the method. Is there any new information?
AC62: Sentence rephrased in L431–434.
RC63. L348: Reanalyzes (verb) => reanalyses (noun)
AC63: Corrected in L438.
RC64. L355, L456: The stratopause variation was not explored at all. How did the authors come to the conclusion on "stratopause disappearance"?
AC64: The stratopause variations are described in new Section 4.3.
RC65. L357: Ambiguous phrase "increase in winter season".
AC65: Deleted.
RC66. L369: What is "the similar processes"? Rather ambiguous.
AC66: Removed.
RC67. L372: "This Z" => "The : : :"
AC67: Removed.
RC68. L378: Who noted this? Could you add a citation?
AC68: Removed.
RC69. L387: Delete "meridian".
AC69: Deleted. The changes in the CO field are illustrated in new Fig. 4, Fig. S1 and S2 and are described in related text.
RC70. L389: Did you show this?
AC70: The results to confirm this are included in new Section 4.1 and are discussed in Section 5.1.
RC71. L400: Did you mean the blocking high?
AC71: Removed. Analysis is focused on the CO re-distribution at different pressure levels in the context of the wave effects (Sections 4.1 and 5.1).
RC72. L404-406: Rephrase this sentence.
AC72: Removed.
RC73. L407: again? How many replacements of warming by cooling happened?
AC73: Rephrased in L522–523.
RC74. L414: What are the large-scale process? The authors seemed to hide some useful information. Another data link appears. The organization of the manuscript needs to be improved.
AC74: Removed. Zonal wave spectra are presented and discussed in new Sections 4.4 and 5.3.
RC75. L419: What do the previous studies refer to? Add the citations directly.
AC75: Citation added in L572–575.
RC76. L435: Add a comma after "(2012)" and after "(2014)".

AC76: This paragraph has been rephrased in L501–509.
RC77. L450: registered? Revised to "documented".
AC77: Corrected, L603.
RC78. L466: Add "is" after "which".
AC78: Corrected in L635.
Proposed references added and discussed.

On behalf of authors
Gennadi Milinevsky
**New text of manuscript:**

[revised manuscript text omitted]

**The description of the data used for analysis**

The Aura MLS CO values have been taken from version 4.2x Aura MLS Level 2 data (https://mls.jpl.nasa.gov/data/readers.php). Aura MLS v4.2x data have 37 pressure levels. The useful range of CO data is from 215 hPa to 0.0046 hPa with corresponding height is from ~11 km to ~86 km. The satellite observation data points are divided into 20º longitude × 2º latitude grids. That means: longitude is divided into 180:20:180 and latitude is divided into 90:2:90 segments. Then the average value of the data is taken in the grid as the value of the center of the grid. For instance, the average in the grid of 180º–160º in longitude and 90º–88º in latitude is taken as the average value of 170º degrees in longitude and 89º in latitude.

Data are removed (replaced by 'NaN') if they do not meet the quality criteria described in 'Version 4.2x Level 2 data quality and description document' (https://mls.jpl.nasa.gov/data/v4-2_data_quality_document.pdf). The five-day average (Fig. S1 and S2) is not simply a sum, divided by five. If the data of a certain area is blank, the data of that area will be ignored on that day. For example, if the data of a certain area in five days are: A, B, NaN, C, NaN, the average value of this area is (A+B+C)/3.

Daily datasets from ERA-Interim global atmospheric reanalysis of European Centre for Medium-Range Weather Forecast have been used for comparison with microwave radiometer observations (https://www.ecmwf.int/en/forecasts/datasets/archive-datasets/reanalysis-datasets/era-interim). Two types of level in the ERA-Interim database were used: 'Model level' and 'Pressure level'. The number vertical levels in 'Model level' and 'Pressure level' datasets are 60 and 37, respectively. The pressure ranges in 'Model level' and 'Pressure level' datasets are from the surface up to 0.1 hPa and 1 hPa, respectively. Horizontal dimension resolution (longitude×latitude) is selected as 0.75°×0.75°. The 'Model type' data are used for drawing temperature and zonal wind velocity profiles from surface up to 0.1 hPa in order to compare with the data measured by microwave radiometer in Kharkiv, which extends up to 87 km altitude. The 'Pressure level' data were used to create geopotential height plots (Fig.1).

**CO movements in stratosphere and mesosphere**

[Figure]

**Figure S1.** The 5-day mean CO fields in the NH stratosphere (0–90°N, between 32 km and 50 km) from the MLS measurements before (first column, 4–8 February), during (second and third columns, 9–13 and 19–23 February, respectively) and after (forth column, 2–6 March) the SSW 2018. White circle shows location of the MWR site Kharkiv relatively the high/low CO amounts marked off by the black contours. Note that Kharkiv falls under the area of high CO amount just after the SSW start (second column, 9–13 February) due to the westward rotation of the polar air mass caused by the zonal wind reverse from westerly to easterly. The high CO anomalies disappear after the SSW (right column, 2–6 March). Dashed lines indicate planetary wave westward tilt with altitude.

[Figure]

**Figure S2.** As in Fig. S1, but for the NH mesosphere (0–90°N, between 64 km and 86 km). Note that the lowest mesospheric CO levels observed with the MWR in February 2018 over Kharkiv (white curve for 6 ppmv in Fig. 3a) are explained by the westward displacement of the boundary between the low- and high-CO polar air mass (compare the Kharkiv location relative to the contour 16 ppmv in (a–c), 6 ppmv in (e–g) and 4 ppmv in (i–k) at 86, 80 and 75 km, respectively. Dashed lines indicate planetary wave westward tilt with altitude.

  **Time–longitude variations and vertical profiles of MLS temperature anomalies**

[Figure]

**Figure S3.** Time–longitude variations of the zonal anomalies in the MLS temperature in the Kharkiv' zone 47.5–52.5°N with respect to the climatology 2005–2017 during January–March 2018. The ten MLS pressure levels between 46 hPa and 0.01 hPa (22 km to 81 km) are presented. Solid (dashed) lines indicate the zonal wave 1 ridge (trough) slowly propagated westward in time and simultaneously displaced westward with altitude. The latter tendency is due to upward propagation of the planetary waves. Dotted lines in the plot for 42 km indicate the change of the zonal anomaly propagation from eastward to westward near 10 February due to zonal wind reverse from westerly to easterly at the start of the SSW 2018.

[Figure]

**Figure S4**. Vertical profiles of the MLS temperature anomalies in January–March 2018 with respect to the mean climatology 2005–2017 over (a) region 47.5–52.5°N, 26–46°E centered at Kharkiv and (b) 47.5–52.5°N zonal mean centered at the Kharkiv latitude. Red vertical lines confine the SSW event 2018.

---

## Author Response (AR2)

**Reply to Referee #1**

Ref: Atmos. Chem. Phys. Discuss., acp-2018-1361, Jun 12, 2019
Title: "Winter 2018 major sudden stratospheric warming impact on midlatitude mesosphere from microwave radiometer measurements" by Yuke Wang et al.

We thank Referee #1 once again for very useful comments, discussion and proposals for corrections. We made corrections and changed the text according to the suggestions. Our revisions and reply to the Referee #1 comments are below in blue colored text.

**Referee Comment: RC**
**Authors Comment: AC**

**RC:** GENERAL COMMENTS

The authors have taken my comments and suggestions carefully into account. Further, they added a wavenumber analysis which, however, does not support the aim of the paper. This is in the interpretation of local mesosphere microwave radiometer observations. So, either the wavenumber analysis of satellite observations is taken out or it is related to the MWR time series. A related major revision should include a careful re-edition.

**RC:** SPECIFIC COMMENTS

There are two options which have to be decided upon: either concentration of the paper (1) or extension (2). In any case, a homogenization (3) is due.

1) Concentration: The authors added with sections 4.4 and 5.3 with figures 7, 8, and 9 a wave analysis of MLS temperatures. This analysis does not help the interpretation of the MWR observations, nor does it forward the knowledge on SSWs. Thats why one option is to erase these additions in order to concentrate the paper.

**AC:** We agree and accept the option (1) proposed by Referee #1 - Concentration - and we corrected the text according this option.

2) Extension: You are presenting wave spectra of MLS temperatures. If it is possible to show a reasonable link to the variability in the MWR carbon monoxide or winds, this would help interpretation of the latter. This could include the fusion of Fig. 8 and 9 for those levels which are shown in Fig. 7. Additionally, a frequency spectrum of MWR wind / carbon monoxide could be shown in another figure. This is the other option.

3) Homogenization: During editing the manuscript, some part of text has doubled and some other technicalities accoured - please, re-edit the text carefully.

**AC:** We removed and modified the text which was doubled. We re-edited the text as well.

**RC:** TECHNICAL CORRECTIONS

L296: "descended up to" --> "descended down to" Corrected, new Line 292.

L302: "explain causes" --> "suggest causes" because you did not quantitatively separate horizontal and vertical transports nor chemical reactions Corrected, L298.

L303: "relative to the pole of the polar vortex" --> "of the polar vortex relative to the pole" Corrected, L300.

L313: "easterly domination" --> "dominance of easterlies" Corrected, L309.

L362: "although" --> "while" Corrected, L357.

L362: In fig. 6a, it is rather the -50-degree-contour than the -55-degree-contour which is descending, or? Corrected, L387.

L395:
"does not accompany by" --> "is not associated with" or "is not accompanied by" Corrected, L390.
"coupling observed" --> "coupling as observed" Corrected, L390.

L422: "transformation" --> "propagation" Text deleted.

L428: End the sentence after "altitude" --> "... altitude. Fig. 8..." Text deleted.

L489: The sentence "The horizontal CO gradient at the polar vortex edge also exists and the vortex split and displacement of the pole associated with the SSW cause significant CO variability at the NH midlatitudes" is highly confusing. Perhaps, it could better express what you mean with "Because of the horizontal CO gradient at the polar vortex edge its split and displacement during the SSW cause a significant CO variability at the NH midlatitudes"? Corrected, L420-421.

L647-L659: This paragraph from "Among the..." to "...split event." is exactly the same as in L470-L481. Please, re-edit this doubling.
AC: The text rephrased and left in Conclusions only.

Fig. 8: The eastward wave 2 should be marked blue in the frequency-wavenumber plot (right column). Fig. 8 removed from the text.

Fig. 9: In the spectra for 20 Dec - 10 Feb (left column) you associated eastward wave 2 with a blue line but do not show it in the spectrum. This wave type is also not indicated in the frequency-wavenumber plots (right column). Fig. 9 removed from the text.

**Reply to Referee #2**

Ref: Atmos. Chem. Phys. Discuss., acp-2018-1361, Jun 12, 2019
Title: "Winter 2018 major sudden stratospheric warming impact on midlatitude mesosphere from microwave radiometer measurements" by Yuke Wang et al.

We thank Referee #2 once again for useful comments, discussion and proposals for corrections. We have included corrections and changes in the text according suggestions. Our revisions and reply to Referee #2 comments are below in blue colored text. RC (AC) is Referee (Authors) comments. Line numbers of the revised manuscript are indicated.

**Referee # 2**

Comments on "Winter 2018 major sudden stratospheric warming impact on midlatitude mesosphere from microwave radiometer measurements" by Yuke Wang, Valeri Shulga, Gennadi Milinevsky, et al. (2019).

**RC:** The general quality of the revised manuscript is greatly improved, compared with the original version. I only have very tiny minor comments this time that should be corrected before publication.

P42: "each two years", better to say "every other year" Corrected, new Line43.
P76: "following" or "followed by"? I guess the latter. Corrected, L80.
P266: Add "a" before "characteristic". Corrected, L282.
P273: What is "the opposite tendency with the stratospheric CO abundance increase"? Why not directly say "A decrease in the stratospheric CO abundance"? **AC:** Corrected to "Unlike the mesosphere, the CO descent and an increase in CO abundance is observed in the stratosphere from both ...", L289.
P276: It is hard for me to understand "descend up". Do you mean "down" rather than "up"? Corrected, L292.
P281: What is "by vertical"? A typo? Corrected, L298-299.
P283: Better to say "shift between A and B" than "shift in A and B"? Corrected, L300.
P293: "Return to" is weird. "Recovery" or "reestablishment" is better. Corrected, L311.
P324: "Positive (negative) values are westerly (easterly) wind components." Please delete the non-informative sentence that says nothing. Deleted
P335: "Return" is weird. Recovery or reestablishment. Corrected, L353.
P337: Move "also" after "is". Corrected, L355.
P353: Add a comma after "i.e." Added, L371.
P358: "looks about"? Do you mean "The stratosphere is …"? Corrected, L377.
P370: "…does not accompany by" should be "… is not accompanied by" Corrected, L390.
P396: Readers will be confused to see "western upper stratosphere".
**AC:** The text deleted according to proposal of Referee 1 - Concentration.
P414: Count once again to see 40-day or 50-day. The text deleted.
P418: "is seen also" should be "is also seen" The text deleted.
P419: "If…then…" is an ill sentence. The text deleted.
P566: "consistent with" rather "consistent to" Corrected, L528.
P619: Change "combine" to "include" Corrected, L553.
P630: A typo is here. It should be "higher …than" rather than "higher … that". Corrected, L565.

**New revised texts are below:**

[revised manuscript text omitted]

**The description of the data used for analysis**

The Aura MLS CO values have been taken from version 4.2x Aura MLS Level 2 data
(https://mls.jpl.nasa.gov/data/readers.php). Aura MLS v4.2x data have 37 pressure levels. The
useful range of CO data is from 215 hPa to 0.0046 hPa with corresponding height is from ~11
km to ~86 km. The satellite observation data points are divided into 20° longitude × 2° latitude
grids. That means: longitude is divided into 180°:20°:180° and latitude is divided into
90°:2°:90° segments. Then the average value of the data is taken in the grid as the value of the
center of the grid. For instance, the average in the grid of 180°–160° in longitude and 90°–88°
in latitude is taken as the average value of 170° degrees in longitude and 89° in latitude.

   Data are removed (replaced by 'NaN') if they do not meet the quality criteria described in
'Version 4.2x Level 2 data quality and description document' (https://mls.jpl.nasa.gov/data/v4-
2_data_quality_document.pdf). The five-day average (Fig. S1 and S2) is not simply a sum,
divided by five. If the data of a certain area is blank, the data of that area will be ignored on that
day. For example, if the data of a certain area in five days are: A, B, NaN, C, NaN, the average
value of this area is (A+B+C)/3.

   Daily datasets from ERA-Interim global atmospheric reanalysis of European Centre for
Medium-Range Weather Forecast have been used for comparison with microwave radiometer
observations (https://www.ecmwf.int/en/forecasts/datasets/archive-datasets/reanalysis-
datasets/era-interim). Two types of level in the ERA-Interim database were used: 'Model level'
and 'Pressure level'. The number vertical levels in 'Model level' and 'Pressure level' datasets
are 60 and 37, respectively. The pressure ranges in 'Model level' and 'Pressure level' datasets
are from the surface up to 0.1 hPa and 1 hPa, respectively. Horizontal dimension resolution
(longitude×latitude) is selected as 0.75°×0.75°. The 'Model type' data are used for drawing
temperature and zonal wind velocity profiles from surface up to 0.1 hPa in order to compare
with the data measured by microwave radiometer in Kharkiv, which extends up to 87 km
altitude. The 'Pressure level' data were used to create geopotential height plots (Fig. 1).

**CO movements in stratosphere and mesosphere**

[Figure]

**Figure S1.** The 5-day mean CO field in the NH stratosphere (0–90°N, between 32 km and 50 km) from the MLS measurements before (first column, 4–8 February), during (second and third columns, 9–13 and 19–23 February, respectively) and after (forth column, 2–6 March) the SSW 2018. White circle shows location of the MWR site Kharkiv relatively the high/low CO amounts marked off by the black contours. Note that Kharkiv falls under the area of high CO amount just after the SSW start (second column, 9–13 February) due to the westward rotation of the polar air mass caused by the zonal wind reverse from westerly to easterly. The high CO anomalies disappear after the SSW (right column, 2–6 March). Dashed lines indicate planetary wave westward tilt with altitude.

[Figure]

**Figure S2.** As in Fig. S1, but for the NH mesosphere (0–90°N, between 64 km and 86 km).
Note that the lowest mesospheric CO levels observed with the MWR in February 2018 over
Kharkiv (white curve for 6 ppmv in Fig. 3a) are explained by the westward displacement of the
boundary between the low- and high-CO polar air mass (compare the Kharkiv location relative
to the contour 16 ppmv in (a–c), 6 ppmv in (e–g) and 4 ppmv in (i–k) at 86, 80 and 75 km,
respectively. Dashed lines indicate planetary wave westward tilt with altitude.

[Figure]

**Figure S3**. Vertical profiles of the MLS temperature anomalies in January–March 2018 with respect to the mean climatology 2005–2017 over (a) region 47.5–52.5°N, 26–46°E centered at

Kharkiv and (b) 47.5–52.5°N zonal mean centered at the Kharkiv latitude. Red vertical lines confine the SSW event 2018.